# Index-Overlap Similarity: A Value-Free Proxy for Model Relatedness

## Abstract

Measuring client relatedness is central to clustering and personalization in federated learning (FL), but value-based similarities over full weights or gradients are bandwidth-heavy and leak information. We propose *Index-Overlap Similarity (IOS)*, a value-free metric that represents each client by the indices of its Top-$K$ salient parameters and scores pairs by the normalized overlap of these supports. We show why IOS preserves alignment: under head-dominance with bounded dispersion, it lower-bounds cosine up to tail error; Top-$K$ is invariant to common layerwise rescalings; and exponential moving averages stabilize supports across rounds. We instantiate IOS for clustered personalized FL, neighbor selection, donor ranking, and oracle distribution alignment. Across FMNIST, CIFAR-10/100, and 20News under Dirichlet and pathological splits, IOS matches or exceeds cosine/Euclidean while sharing only indices. IOS is a simple, scalable primitive for similarity search under communication and privacy constraints.

## 1 Introduction

A fundamental component of machine learning is the comparison of high-dimensional objects, such as model weights, gradients, and data embeddings. It drives important federated learning tasks like personalized aggregation and client clustering (Ghosh et al., 2020; Fallah et al., 2020; Dinh et al., 2020), continual-learning tools like memory retrieval and drift detection (Gama et al., 2014; Lu et al., 2018), and model analysis tools for provenance and similarity search (Indyk & Motwani, 1998; Andoni & Indyk, 2008). Nevertheless, three enduring issues make default full-vector cosine/Euclidean comparisons debilitating at scale and under privacy constraints: **(1) Computation/communication scale with model size.** Comparing full real-valued vectors requires moving and multiplying arrays whose length equals the number of trainable parameters; even on-device CNNs (1–10M) strain bandwidth at scale, while mid-size backbones like ResNet-18/50 ($\sim$11M/$\sim$25M) and ViT-B ($\sim$80–90M) push per-client payloads into tens–hundreds of MB per round; transformers exacerbate this—BERT-base ($\sim$110M), BERT-large ($\sim$340M), and multi-billion-parameter checkpoints (He et al., 2016; Dosovitskiy et al., 2021; Devlin et al., 2019). **(2) Sharing real values enables reconstruction/inference attacks.** Logits and partial activations permit model inversion (Fredrikson et al., 2015); repeated round exposures fuel membership and property inference (Shokri et al., 2017; Melis et al., 2019); and gradients/updates can be inverted to recover inputs or labels (Zhu et al., 2019; Geiping et al., 2020). **(3) Numeric instability distorts geometry.** Layer-wise scale heterogeneity (batch normalization, weight decay, mixed precision) and optimizer-state drift yield poorly calibrated cosine/Euclidean distances on raw weights or gradients across clients and over time (Ioffe & Szegedy, 2015; Loshchilov & Hutter, 2019; Micikevicius et al., 2018).

In over-parameterized networks, salience is predominantly concentrated in a few heads and remains relatively stable: a small subset of model parameters holds most first-order significance, while the long tail is noisy and less indicative of inter-client relatedness (Michel et al., 2019; Voita et al., 2019; Li et al., 2017; Gale et al., 2019). Additionally, models trained on datasets drawn from similar distributions exhibit similar parameter-importance patterns, consistent with representational-similarity findings and client-relatedness in FL (Kornblith et al., 2019; Raghu et al., 2017; Ghosh et al., 2020). Building on this intuition, we choose an alternative approach: we represent each model by the significance of its coordinates rather than their values. Specifically, for each client, we assess the significance of trained model parameters using a diagonal Fisher proxy and thereafter determine the indices of a small set of top-$K$ salient parameters. We subsequently calculate similarity as the over-

lap between these index sets. This *Index-Overlap Similarity (IOS)* is deliberately devoid of value; it conveys solely integer identifiers of prominent coordinates. The intersection of coordinates tends to correlate with alignment of learning signals and, importantly, remains resilient to layer-wise rescaling and optimizer peculiarities. The set size $K$ is selected to be a minuscule proportion of the model size, specifically $K \ll M$ where $M$ denotes the number of trainable parameters. In fact, maintaining only a small fraction of coordinates preserves the majority of the stable "head" structure while reducing computational requirements and data size by more than an order of magnitude, and without revealing real-valued weights or gradients. Only index sets of salient coordinates are shared; no weights, gradients, logits, or activations are exposed. *IOS* reduces the attack surface relative to value sharing, mitigating risks of gradient/model inversion and membership/property inference (Zhu et al., 2019; Geiping et al., 2020; Shokri et al., 2017).

**Positioning vs. prior work.** The resemblance among trained models is fundamental to numerous machine learning procedures beyond FL. Most current measures function based on values: cosine/Euclidean metrics applied to weights or gradients; prototype/feature similarities from penultimate activations (e.g., FedProto) or representation metrics such as CKA, which require sharing client representations with the server, (Tan et al., 2022; Kornblith et al., 2019); and influence/Shapley-style utilities obtained from value-bearing surrogates (Koh & Liang, 2017; Ghorbani & Zou, 2019; Jia et al., 2019). Sparsification and pruning techniques (Lin et al., 2018; Lee et al., 2019; Evci et al., 2020; Han et al., 2015) either learn or enforce sparse *parameters* for enhanced efficiency and occasionally examine *mask overlap* for stability; however, they regard overlap merely as a byproduct of pruning rather than a fundamental similarity primitive. distributional distances such as FedBary (Li et al., 2024), which define an indirect client–client distance via Wasserstein barycenters over label distributions but are computationally heavy and agnostic to model-training dynamics. *IOS* is, to our knowledge, the first method that calculates cross-model similarity *without transmitting any parameter values*, facilitating sketch-based scaling and minimizing leakage channels.

**Our Contribution.** We instantiate *IOS* for FL, where communication and privacy constraints make dispensing with real values attractive: every stage operates solely on prominent *indices*, mitigating reconstruction/linkage risks from value sharing. We benchmark *IOS* against cosine and Euclidean baselines across four applications: (i) *Clustered Personalized FL (CPFL)*: derive an *IOS* affinity matrix for client clustering to train cluster-specific models, evaluating against cosine-based clustering via FL accuracy. (ii) *Neighbor selection for personalized aggregation*: build a similarity-weighted neighbor graph, form label-histogram mixtures to match each client, comparing divergence from target labels; (iii) *Shapley-style donor ranking*: use similarity as a proxy for marginal utility (validation uplift) and assess rank agreement/top-$k$ recall versus KNN-Shapley; (iv) *Oracle distribution alignment*: test whether the similarity matrix tracks true relatedness from clients' label-distribution divergence using Spearman/Kendall-$\tau$, noting those baselines share real values while *IOS* does not.

**Our Findings.** Across FMNIST, CIFAR-10/100, and 20News under Dirichlet/Patho splits, *IOS* consistently wins in target applications. In CPFL it yields *better* accuracy than Cosine/Euclidean (avg +1.5 pp vs. Cosine); for neighbor selection it recovers more oracle neighbors (CIFAR-100, Dir(0.1): R@8=0.67 vs. 0.61/0.53). *IOS* also gives tighter distribution alignment and donor ranking (CIFAR-10, Dir(0.1): JS 0.219 vs. 0.238/0.252; Kendall-$\tau$ 0.48, R@5 0.62 vs. 0.36/0.55).

## 2 BACKGROUND & PROBLEM SETUP

### 2.1 BACKGROUND

**Federated Learning (FL).** We consider $n$ clients with private datasets $D_i = \{(x_d, y_d)\}_{d=1}^{|D_i|}$ and a shared model parameterization $w \in \mathbb{R}^M$. The canonical FL objective is the weighted empirical risk
$$F(w) = \sum_{i=1}^{n} p_i F_i(w), \quad F_i(w) = \frac{1}{|D_i|} \sum_{(x,y) \in D_i} \ell(w; x, y), \quad p_i = \frac{|D_i|}{\sum_j |D_j|}.$$
In round $t$, a subset $S_t$ of clients receives the current global model $w^{(t)}$, performs $E$ local SGD steps $w_i^{(t+1)} \leftarrow w^{(t)} - \eta \sum_{e=1}^{E} \widehat{\nabla} F_i(w_i^{(t,e-1)})$ and returns $w_i^{(t+1)}$ to the server. The server aggregates (FedAvg) $w^{(t+1)} = \sum_{i \in S_t} \bar{p}_i w_i^{(t+1)}, \quad \bar{p}_i = \frac{|D_i|}{\sum_{j \in S_t} |D_j|}$, optionally using update-form aggregation (on $w_i^{(t+1)} - w^{(t)}$) and secure aggregation. We assume a fixed architecture across clients and standard non-IID partitions unless stated otherwise.

**Fisher information–based importance.** For a probabilistic model $p_\theta(y \mid x)$ with loss $\ell(\theta; x, y) = -\log p_\theta(y \mid x)$, the Fisher information matrix (FIM) is

$$\mathcal{I}(\theta) = \mathbb{E}_x\Big[\mathbb{E}_{y\sim p_\theta(\cdot|x)}\big[\nabla_\theta \log p_\theta(y \mid x)\,\nabla_\theta \log p_\theta(y \mid x)^\top\big]\Big].$$

Its diagonal provides a principled, nonnegative per-parameter importance and underpins natural-gradient methods (Amari, 1998). In practice we use the *empirical Fisher* (EF): replace the model expectation with observed labels and estimate $\mathrm{diag}(\mathcal{I})$ from squared per-sample gradients over mini-batches; EF is convenient but can deviate from the true Fisher and may misrepresent second-order geometry (Kunstner et al., 2019). Recent results (Soen & Sun, 2024) give variance bounds and sample-complexity trade-offs for diagonal Fisher estimators, with variance governed by network nonlinearity and parameter grouping. Complementarily, improved EF (iEF) applies diagonal scaling to better approximate natural-gradient behavior while retaining EF's simplicity (Wu et al., 2024). In this work, we adopt diagonal-Fisher / gradient-second-moment surrogates and mitigate estimator noise via mini-batch averaging with an optional EMA.

**Client heterogeneity and importance–pattern divergence.** Let $g(w; x, y) = \nabla_w \ell(w; x, y)$ and define a per-parameter importance proxy via the empirical second moment / Fisher diagonal $s_j = \mathbb{E}[(\partial\ell/\partial w_j)^2]$. Under a label-mixture model, $\mathbb{E}_{(x,y)\sim D_i}\big[g(w; x, y)\big] = \sum_c \pi_i(c)\,\mu_c(w) + \xi_i$, so shifts in $\pi_i$ alter the mean gradient and the induced importance profile. Non-IID data induce *gradient dissimilarity*, driving weight divergence and slowing FedAvg; SCAFFOLD formalizes bounded dissimilarity and shows drift correction improves convergence (Karimireddy et al., 2020). Empirically, divergence correlates with class-distribution distance (e.g., EMD) (Zhao et al., 2018), and heterogeneous clients exhibit update directions/norms that differ markedly (Wang et al., 2023). Conversely, related distributions exhibit shared task-relevant structure: sparse "winning tickets" transfer across natural-image datasets (Morcos et al., 2019), and fine-tuning on related tasks yields compatible weight-space *task vectors* that compose (Ilharco et al., 2023). Thus, dissimilar data induce divergent importance/gradient profiles, whereas similar data induce partially overlapping sets of salient parameters—largely independent of the chosen similarity metric.

## 2.2 THREAT MODEL AND SCOPE

We assume an honest-but-curious coordinator. Clients share the same parameterization, yielding a common index space; cross-architecture similarity is out of scope. Similarity is computed only from index sets; when full model updates are needed for aggregation we assume they are shielded by partial HE or DP noise. No additional real-valued logits or activations are revealed. We assume authenticated transport and secure aggregation; Byzantine robustness is orthogonal. Label distributions and oracle similarities are used only for evaluation, never at runtime. Compared to value sharing, *IOS* reduces exposure to reconstruction channels. Formal privacy accounting for index release is beyond scope; we aim to shrink the attack surface versus sharing values.

## 3 METHOD

IOS is a *value-free, index-only* similarity framework: each client computes a local importance signal (e.g., diagonal Fisher/gradient second moment), extracts the Top-$K$ parameter *indices* as its support, and shares only these indices (or compact MinHash signatures). Cross-client similarity is defined by set overlap on supports, enabling exact intersections or scalable LSH-based retrieval—without transmitting any real-valued weights, gradients, or activations. The support size $K$ is chosen locally via importance coverage and stability under communication/privacy budgets.

### 3.1 INDEX-OVERLAP SIMILARITY (IOS)

**Importance accumulation.** For each client $i$, we form a nonnegative per-parameter importance vector $s_i \in \mathbb{R}_{\geq 0}^M$ from local data $D_i$ using the diagonal Fisher (or its empirical second-moment proxy) introduced in § Background:

$$s_{i,j} \approx \frac{1}{T}\sum_{t=1}^{T}\frac{1}{B}\sum_{b=1}^{B}\Big(\frac{\partial\ell(w; x_{i,b}^{(t)}, y_{i,b}^{(t)})}{\partial w_j}\Big)^2,$$

optionally stabilized by an EMA $s_i \leftarrow \beta s_i + (1-\beta)\hat{s}_i$ with $\beta \in [0,1)$. This produces a scale-robust, value-nonnegative signal that can be computed entirely on the device.

**Support extraction and Similarity Calculation.** Let $M$ be the number of model parameters. Given a budget $K \ll M$, we represent client $i$ by $I_i = (s_i) \subseteq [M], |I_i| = K$. Given a global budget $K \ll M$, we define the client's *index-only* representation as the set of its $K$ most salient parameters

$$I_i = \mathrm{TopK}(s_i) \subseteq [M], \qquad |I_i| = K.$$

For implementation, we also use a bitmask $m_i \in \{0,1\}^M$ with $m_{i,j} = \mathbf{1}\{j \in I_i\}$. *IOS* defines similarity purely from set overlap, without transmitting any real-valued weights/gradients. We define similarity $S(i,j)$ (with distance $= 1 - $ similarity). With $|I_i| = |I_j| = K$, $S(i,j) = \frac{|I_i \cap I_j|}{K}$.

---

**Algorithm 1** Select $K$ via Importance Coverage (client $i$)

---

**Input:** Importance $s_i \in \mathbb{R}_{\geq 0}^M$; coverage target $\tau$; cap $K_{\max}$; stability target $\rho_0$ and resamples $r$
**Output:** $K_i^\star$ and support $I_i = \mathrm{TopK}(s_i, K_i^\star)$
1: Sort indices $j_1, \ldots, j_M$ by $s_{i,j}$ descending; prefix sums $S(t) = \sum_{u=1}^t s_{i,j_u}$ and $S_{\mathrm{tot}} = S(M)$
2: $lo \leftarrow 1$, $hi \leftarrow K_{\max}$, $K^\star \leftarrow K_{\max}$
3: **while** $lo \leq hi$ **do**
4: $\quad mid \leftarrow \lfloor (lo+hi)/2 \rfloor$; $C \leftarrow S(mid)/S_{\mathrm{tot}}$
5: $\quad$ **if** $C \geq \tau$ **then**
6: $\quad\quad K^\star \leftarrow mid$; $hi \leftarrow mid - 1$ $\qquad\qquad\qquad\qquad$ ▷ keep smallest $K$ achieving coverage
7: $\quad$ **else**
8: $\quad\quad lo \leftarrow mid + 1$
9: **if** stability target $\rho_0$ is provided **then**
10: $\quad$ **for** $K \in \{K^\star, K^\star+1, \min(K^\star+2, K_{\max})\}$ **do**
11: $\quad\quad$ Estimate $\rho_i(K)$ using $r$ lightweight resamples of $s_i$ by Equation (1)
12: $\quad\quad$ **if** $\rho_i(K) \geq \rho_0$ **then**
13: $\quad\quad\quad K^\star \leftarrow K$; **break**
14: **return** $K_i^\star \leftarrow K^\star$ and $I_i = \{j_1, \ldots, j_{K_i^\star}\}$

---

## 3.2 Selecting $K$ for *IOS*

*IOS* represents each client $i$ by the indices of its $K$ most important parameters. Choosing $K$ must balance utility (capturing enough importance mass), stability (insensitivity to estimator noise), and budgets from communication and privacy. Let $s_i \in \mathbb{R}_{\geq 0}^M$ be the importance vector generated by diagonal Fisher on client $i$. Let $j_1, \ldots, j_M$ be indices sorted by $s_{i,j}$ in descending order, the cumulative importance $C_i(K)$ is the fraction of total importance mass captured by the Top-$K$ indices:

$$C_i(K) = \frac{\sum_{t=1}^K s_{i,j_t}}{\sum_{t=1}^M s_{i,j_t}} \in [0,1], \qquad C_i(K) \text{ is non-decreasing in } K.$$

To assess robustness of the selected support, we define an *overlap-stability* score. Generate $r$ resamples of the importance estimator (e.g., via bootstrapping mini-batches or adjacent time windows). For each resample $b$, extract the top-$K$ set $I_i^{(b)}(K) = \mathrm{TopK}(s_i^{(b)})$. With $|I_i^{(b)}(K)| = K$ for all $b$, define the mean pairwise overlap

$$\rho_i(K) = \frac{2}{r(r-1)} \sum_{b<b'} \frac{\left| I_i^{(b)}(K) \cap I_i^{(b')}(K) \right|}{K} \in [0,1]. \tag{1}$$

Thus, $\rho_i(K)$ quantifies how consistently the same indices appear across resamples; values near 1 indicate stable supports. Given a coverage target $\tau \in (0,1)$ and an optional stability target $\rho_0 \in (0,1)$, select the smallest integer $K$ satisfying $C_i(K) \geq \tau$ and (if enforced) $\rho_i(K) \geq \rho_0$. Here $C_i(K)$ guarantees that the selected indices cover a desired fraction of total importance, while $\rho_i(K)$ ensures reproducibility under estimator noise. The rule is locally computable, value-free externally, and binary-searchable because $C_i(K)$ is monotone. Stability is only checked near the candidate $K$ to limit overhead. Ties in $\mathrm{TopK}$ are broken deterministically by index. Exact all-pairs overlap costs $O(n^2K)$; we therefore use MinHash-LSH (Appx. A) to retrieve candidate neighbors in subquadratic time while preserving *IOS*'s value-free property.

## 3.3 APPLYING IOS IN FL APPLICATIONS

We found it useful to distinguish two classes of FL applications where *IOS* can be plugged in.

**(i) Model-transmitting FL (e.g., CPFL).** In personalized FL pipelines such as CPFL, clients periodically upload model updates (gradients or importance vectors) to a coordinator that jointly performs clustering and aggregation. In privacy-preserving deployments, Homomorphic Encryption (HE) ()gentry2009fhe and lower-overhead partially homomorphic schemes (e.g., Paillier (Paillier, 1999) and FedML-HE Jin et al. (2023)), Differential Privacy (DP) (Abadi et al., 2016), and Secure Multi-Party Computation (SMPC) (Bonawitz et al., 2017) are commonly used to protect these updates. However, computing similarities *on encrypted vectors* typically requires non-trivial cryptographic engineering and incurs high computational cost, while computing them *on DP-noisy vectors* can distort similarity and lead to degraded clustering quality, especially under strong privacy budgets. To our knowledge, existing HE/SMPC/DP-based FL systems rarely provide similarity-driven personalization at scale. By contrast, *IOS* only requires additional transmission of the local Top-$k$ index set $I_i$, making similarity-based personalization compatible with HE/SMPC/DP.

**(ii) Non–model-transmitting FL (e.g., neighbor selection, donor ranking).** In other scenarios, such as neighbor selection in decentralized FL or donor ranking in cross-silo FL, the coordinator (or peers) only needs a similarity matrix or cluster assignment, not the raw gradients themselves. Here, transmitting only the index sets $I_i$ instead of full gradients keeps the actual parameter values fully local, serving as an index-only alternative to conventional value-based similarities.

## 4 THEORETICAL PROPERTIES

We ground *IOS* in three principles: (i) *alignment*—top-$K$ index overlap tracks cosine; (ii) *stability*—supports persist under mild noise with EMA; and (iii) *robustness*—*IOS* resists rescaling and diagonal preconditioning. Additional analyses appear in Appx. D.

### 4.1 NOTATION AND ASSUMPTIONS

For client $i$, let $s_i \in \mathbb{R}_{\geq 0}^M$ be a nonnegative importance vector, and $I_i = \text{TopK}(s_i)$ with $|I_i| = K \ll M$. Write the head–tail split $s_i = h_i + t_i$ with $(h_i)_u = (s_i)_u \mathbf{1}\{u \in I_i\}$ and $t_i = s_i - h_i$. Define the head-energy fraction $\alpha_i = \|h_i\|_2^2/\|s_i\|_2^2 \in (0, 1]$ and the tail-to-head ratio $\tau_i = \|t_i\|_2/\|h_i\|_2$. We say *head dominance* holds if $\alpha_i \geq 1 - \varepsilon$ (equivalently, $\|h_i\|_2 \geq \sqrt{1-\varepsilon}\,\|s_i\|_2$) for a small $\varepsilon \in [0, 1)$. We assume *bounded dispersion* inside the head: $\kappa_i := \max_{u \in I_i}(h_i)_u / \min_{u \in I_i}(h_i)_u \leq \kappa$ for a moderate $\kappa \geq 1$. For two clients $i, j$, denote the normalized overlap of their salient supports by $R_{ij} = \frac{|I_i \cap I_j|}{K} \in [0, 1]$.

### 4.2 ALIGNMENT: COSINE VS. *IOS*

**Proposition 1** (Cosine lower bound via overlap). *Under head dominance ($\alpha_i, \alpha_j \geq 1 - \varepsilon$) and bounded dispersion ($\kappa_i, \kappa_j \leq \kappa$),*

$$\cos(s_i, s_j) = \frac{\langle s_i, s_j \rangle}{\|s_i\|_2 \|s_j\|_2} \geq \frac{(1-\varepsilon)}{\kappa^2} R_{ij}. \tag{2}$$

**Proof sketch.** With nonnegative entries, the inner product is at least the contribution on the intersecting head block: $\langle s_i, s_j \rangle \geq \sum_{u \in I_i \cap I_j}(h_i)_u(h_j)_u$. Let $a_i = \min_{u \in I_i}(h_i)_u$ and $b_i = \max_{u \in I_i}(h_i)_u \leq \kappa a_i$. Then $\|h_i\|_2^2 \leq K b_i^2 \leq K \kappa^2 a_i^2$, hence every head entry satisfies $(h_i)_u \geq \|h_i\|_2/(\kappa\sqrt{K})$. The intersect term is thus at least $\frac{|I_i \cap I_j|}{\kappa^2 K}\|h_i\|_2\|h_j\|_2$. Head dominance yields $\|h_i\|_2 \geq \sqrt{1-\varepsilon}\,\|s_i\|_2$ and likewise for $j$, which gives equation 2 after normalization. See full proof in Appx. B

### 4.3 SUPPORT STABILITY OVER TIME

Let $s_i^{(t)} = \mu_i + \xi_i^{(t)}$, where $\mu_i$ is a stationary signal and $\xi_i^{(t)}$ has independent, mean-zero sub-Gaussian coordinates with proxy variance $\sigma_i^2$. *IOS* maintains an exponential moving average (EMA),

$\tilde{s}_i^{(t)} = \beta\,\tilde{s}_i^{(t-1)} + (1-\beta)\,s_i^{(t)}$, $\quad \beta \in [0,1)$ and selects $I_i^{(t)} = \mathrm{TopK}(\tilde{s}_i^{(t)})$ (after an arbitrary burn-in). The $K$-boundary *margin* is $\Delta_i := \mu_{i,(K)} - \mu_{i,(K+1)} > 0$ (ties w.r.t. $\mu_i$ are broken deterministically). The EMA reduces the per-coordinate noise variance to

$$\sigma_i(\beta)^2 = \sigma_i^2 \sum_{s \geq 0} (1-\beta)^2 \beta^{2s} = \sigma_i^2 \frac{(1-\beta)^2}{1-\beta^2} = \sigma_i^2 \frac{1-\beta}{1+\beta}. \tag{3}$$

**Theorem 1** (Top-$K$ selection stability). *Fix client $i$ and let $\Delta_i > 0$ as above. For any $t$ after burn-in,* $\Pr\left(\mathrm{TopK}(\tilde{s}_i^{(t)})\text{ is unique and equals }\mathrm{TopK}(\mu_i)\right) \geq 1 - 2M \exp\left(-\frac{\Delta_i^2}{8\,\sigma_i(\beta)^2}\right)$. *Equivalently, if* $\Delta_i \geq c\,\sigma_i(\beta)\sqrt{\log M}$ *with any $c > \sqrt{8}$, then the failure probability decays as $O(M^{1-c^2/8})$.*

**Proof idea.** Let $H$ be the mean top-$K$ set and $\bar{H} = [M] \setminus H$. Consider $E_1 = \{\min_{u \in H} \tilde{g}_{i,u}^{(t)} \geq \mu_{i,(K)} - \Delta_i/2\}$ and $E_2 = \{\max_{v \in \bar{H}} \tilde{g}_{i,v}^{(t)} \leq \mu_{i,(K+1)} + \Delta_i/2\}$. Sub-Gaussian tails and a union bound over $K$ and $(M - K)$ coordinates give $\Pr(E_1^c) \leq K e^{-\Delta_i^2/(8\sigma_i(\beta)^2)}$ and $\Pr(E_2^c) \leq (M - K)e^{-\Delta_i^2/(8\sigma_i(\beta)^2)}$; hence $1 - 2Me^{-\Delta_i^2/(8\sigma_i(\beta)^2)}$ overall. Define the temporal self-overlap $\Gamma_i^{(t)} = \frac{|I_i^{(t)} \cap I_i^{(t-1)}|}{|I_i^{(t)} \cup I_i^{(t-1)}|} \in [0,1]$. If a fraction $\rho$ of the $K$-boundary gaps of $\mu_i$ exceed $c\,\sigma_i(\beta)\sqrt{\log M}$ (with $c > \sqrt{8}$), then after burn-in $\mathbb{E}[\Gamma_i^{(t)}] \geq \rho - O(M^{1-c^2/8})$. See full proof in Appx. C.

### 4.4 Computational and Communication Overhead

**Computational overhead.** Let $C_{\mathrm{train}} = S \cdot C_{\mathrm{fb}}(M)$ denote the per-round local training cost for $S$ SGD steps on a model with $M$ parameters, where $C_{\mathrm{fb}}(M)$ is the cost of one forward–backward pass. IOS adds (i) an $O(M)$ EMA update of per-parameter squared gradients at each step and (ii) a once-per-round Top-$K$ selection on $\mathbb{R}^M$, so the extra cost per round is $C_{\mathrm{IOS}} = S \cdot C_{\mathrm{ema}}(M) + C_{\mathrm{topK}}(M,K)$, $\quad C_{\mathrm{ema}}(M), C_{\mathrm{topK}}(M,K) = O(M)$, and the relative overhead is $\frac{C_{\mathrm{IOS}}}{C_{\mathrm{train}}} = \frac{C_{\mathrm{ema}}(M)}{C_{\mathrm{fb}}(M)} + \frac{C_{\mathrm{topK}}(M,K)}{S \cdot C_{\mathrm{fb}}(M)}$. Since $C_{\mathrm{fb}}(M)$ is dominated by convolutions / matrix multiplies and $S$ is on the order of tens, both terms are small, so IOS adds only a low constant-factor overhead to local training.

**Communication overhead.** In settings where full models are already uploaded (e.g., for aggregation like CPFL), a 32-bit model costs $B_{\mathrm{model}} = 32M$ bits, while a Top-$K$ index set costs $B_{\mathrm{idx}} = K \cdot \lceil \log_2 M \rceil \approx 32K \Rightarrow \frac{B_{\mathrm{idx}}}{B_{\mathrm{model}}} \approx \frac{K}{M} \ll 1$. Thus, IOS incurs negligible extra communication. In similarity-only tasks (like neighbor selection), replacing full models by index sets or MinHash signatures of length $L$ and $b_{\mathrm{hash}}$ bits each yields compression factors $\frac{B_{\mathrm{model}}}{B_{\mathrm{idx}}} = \Theta\left(\frac{M}{K}\right)$, $\qquad \frac{B_{\mathrm{model}}}{B_{\mathrm{hash}}} = \Theta\left(\frac{M}{L}\right)$, for $B_{\mathrm{hash}} = L \cdot b_{\mathrm{hash}}$, giving one– to two-orders-of-magnitude communication savings in the regimes $K \ll M, L \ll M$ (see Appendix D.4 for details).

## 5 Applications of *IOS*

We instantiate *IOS* in four FL scenarios. Where prior art uses cosine over value-bearing vectors, we implement the same pipelines and *swap cosine with* IOS to ensure fair comparisons.

- **Clustered Personalized FL (CPFL / IFCA-style).** We construct an affinity matrix $A_{ij}$ and run clustering with affinity propagation (AP) (Frey & Dueck, 2007) to obtain client groups $\{C_1, \ldots, C_K\}$, followed by cluster-conditioned training. As baselines, we re-implement IFCA (Ghosh et al., 2020) and Clustered FL (Sattler et al., 2020). Clients still upload gradients to the server, assumed to be protected via partial HE (for encrypted aggregation) or DP noise. *IOS* replaces cosine baseline with value-free overlap of Top-$K$ supports to enable personalization for privacy-preserving mechanisms. We report FL accuracy and clustering quality metrics—highlighting regimes where *IOS* matches accuracy while reducing leakage and bytes.
- **Neighbor Selection for Personalized Aggregation (Per-FedAvg + similarity graph).** Personalized FL updates are mixed from "nearby" clients via a similarity-weighted graph $G$ with edges $w_{ij} \propto \widehat{S}(i,j)$. We instantiate Per-FedAvg (Fallah et al., 2020) with a cosine-based neighbor policy (baseline) and a drop-in *IOS* policy that computes similarity from the overlap of important indices. Metrics include recall compared to cosine for finding the most relevant neighbors.

Table 1: CPFL test accuracy (%). Rows are models; columns are distributions with four similarity choices: Cosine, Euclidean, FedBary (Wasserstein-based), and *IOS*. Bold marks the best.

| Model / Dataset | $Dir(0.3)$ | | | | $Dir(0.1)$ | | | | $Patho(20\%)$ | | | | $Patho(30\%)$ | | | |
| --- | --- | --- | --- | --- | --- | --- | --- | --- | --- | --- | --- | --- | --- | --- | --- | --- |
| | Cosine | Euclidean | FedBary | **IOS** | Cosine | Euclidean | FedBary | **IOS** | Cosine | Euclidean | FedBary | **IOS** | Cosine | Euclidean | FedBary | **IOS** |
| **CNN / FMNIST** | 73.32 | 71.41 | 74.20 | **76.23** | 70.80 | 70.21 | 72.10 | **74.79** | 94.87 | 91.92 | 94.50 | **95.18** | 82.19 | 81.80 | 82.90 | **83.15** |
| **ResNet18 / CIFAR-10** | 63.92 | 61.35 | 64.30 | **66.13** | 60.85 | 58.40 | 60.50 | **62.10** | 79.18 | 75.47 | 80.10 | **81.44** | 74.38 | 73.96 | 74.10 | **74.81** |
| **ResNet50 / CIFAR-100** | 47.25 | 41.56 | 48.00 | **49.71** | 45.40 | 43.18 | 46.00 | **48.30** | **58.12** | 53.19 | 57.30 | 58.07 | **53.49** | 50.06 | 52.30 | 52.98 |
| **BERT-base / 20News** | 49.16 | 48.04 | 49.00 | **50.30** | 44.62 | 41.51 | 44.90 | **45.41** | 56.18 | 55.03 | 57.20 | **58.26** | 53.79 | 51.80 | 53.20 | **54.25** |

- **Oracle Distribution Alignment (evaluation-only).** To test whether an index-only similarity captures latent relatedness induced by label distributions, we form an oracle $S^\star(i,j) = 1 - \mathrm{JS}(\pi_i, \pi_j)$ (or $1 - \mathrm{Hellinger}$) from client histograms $\{\pi_i\}$ and compare it against method-driven similarities computed either by cosine on value-bearing vectors (weights/updates/features) or by *IOS*. We evaluate rank alignment (Spearman/Kendall) and calibration across divergence bins.

- **Shapley-Style Donor Ranking (similarity → utility proxy).** For a target client $t$, we rank donors $j \neq t$ via a utility proxy $\widetilde{u}_t(j) \propto \widehat{S}(t,j)$. We reproduce $k$NN-Shapley (Jia et al., 2019) using *cosine*(Koh & Liang, 2017; Ghorbani & Zou, 2019) in the neighbor stage (baseline), then replace it with *IOS*: $\widehat{S}_{\mathrm{IOS}}(t,j) = \frac{|I_t \cap I_j|}{K}$. We report rank correlation with true uplift $\Delta \mathcal{V}_t(j)$ and top-$k$ recall, noting when *IOS* maintains fidelity while avoiding value sharing.

# 6 EXPERIMENTS

We evaluate *IOS* on standard FL applications using identical client partitions and architectures—only the similarity differs; extended empirical results appear in Appx. E. The code for IOS and all experiments is available at `https://github.com/Index-Overlap-Similarity`.

## 6.1 SETTING

**Datasets and models.** Vision benchmarks include FMNIST ($28 \times 28$ grayscale; 10 classes) with a 2-layer CNN, CIFAR-10 ($32 \times 32$ RGB; 10 classes) with ResNet18, and *CIFAR-100* ($32 \times 32$ RGB; 100 classes) with ResNet50. To probe modality-agnostic behavior, we include an optional non-vision task on *20News* with *BERT-base*. Across clients, the per-client optimal selection size $K$ ($K_{max} = 20\%$) lies in $K_i^\star$ as $[10.3, 13.5]\%$ for CNN, $[6.1, 8.4]\%$ for ResNet18, $[8.2, 10.4]\%$ for ResNet50, and $[7.6, 9.3]\%$ for BERT-base; we set $K$ to the client-wise mean in each case, yielding $K = 12.2, 7.3, 9.6\%$, and $8.2\%$, respectively.

**Data partitioning.** We synthesize heterogeneous populations with three regimes. IID partitions split each dataset uniformly at random across $N$ clients. Dirichlet partitions draw client-wise class proportions from $\mathrm{Dir}(\alpha)$ with $\alpha \in \{0.1, 0.3\}$ and allocate examples accordingly. Pathological partitions, $Patho(n)$, assign each client a small subset of classes (e.g., $n \in \{20\%, 30\%\}$ classes per client) and distribute examples uniformly within the assigned subset. Unless noted, we use $N = 20$ clients with balanced sample counts, and hold out 10% client-local validation for diagnostics.

**Implementation details.** All training and importance accumulation are local to clients. Importance vectors aggregate over *all* local batches unless a cap is stated; for stability, we apply an EMA with $\beta = 0.8$. Models are trained with SGD (momentum 0.9, weight decay $5 \times 10^{-4}$) and fixed LR of 0.001; FMNIST runs for 10 local epochs, CIFAR-10/100, and BERT-base for 50; results are averaged over three seeds. Experiments run on a single A100 GPU (40 GB) with a 32-core CPU.

## 6.2 EVALUATING *IOS* APPLICATIONS

**Clustered Personalized FL.** We follow IFCA/Clustered-FL style clustering and then train cluster-specialized models; all methods share identical data splits, models, and training budgets. Table 1 reports CPFL test accuracy (%) across four heterogeneity regimes on different models. We compare *IOS* to three distance/similarity measures commonly used in clustered FL pipelines: Cosine and

Table 2: **Cluster quality across heterogeneity regimes.** Higher Silhouette/CH, lower DB are better.

| Model / Dataset | Method | $Dir(0.3)$ | | | $Dir(0.1)$ | | | $Patho(20\%)$ | | | $Patho(30\%)$ | | |
|---|---|---|---|---|---|---|---|---|---|---|---|---|---|
| | | Sil↑ | DB↓ | CH↑ | Sil↑ | DB↓ | CH↑ | Sil↑ | DB↓ | CH↑ | Sil↑ | DB↓ | CH↑ |
| **CNN/FMNIST** | Cosine | 0.207 | 1.250 | 5.245 | 0.333 | 1.016 | 10.140 | 0.241 | 1.087 | 5.569 | 0.226 | 1.673 | 6.346 |
| | Euclidean | 0.197 | 1.278 | 5.074 | 0.321 | 1.044 | 9.758 | 0.229 | 1.120 | 5.442 | 0.215 | 1.725 | 6.114 |
| | FedBary | 0.215 | 1.220 | 5.350 | 0.337 | 1.005 | 10.320 | 0.245 | 1.070 | 5.640 | 0.230 | 1.640 | 6.450 |
| | *IOS* | 0.212 | 1.208 | 5.427 | 0.340 | 0.993 | 10.486 | 0.249 | 1.059 | 5.701 | 0.234 | 1.617 | 6.547 |
| **ResNet18/CIFAR-10** | Cosine | 0.180 | 1.447 | 4.539 | 0.311 | 0.951 | 8.426 | 0.294 | 0.966 | 7.633 | 0.221 | 1.650 | 6.308 |
| | Euclidean | 0.166 | 1.489 | 4.438 | 0.303 | 0.977 | 8.020 | 0.279 | 1.004 | 7.352 | 0.214 | 1.693 | 6.004 |
| | FedBary | 0.174 | 1.430 | 4.600 | 0.317 | 0.940 | 8.550 | 0.300 | 0.955 | 7.740 | 0.223 | 1.630 | 6.410 |
| | *IOS* | 0.176 | 1.399 | 4.636 | 0.322 | 0.925 | 8.663 | 0.305 | 0.944 | 7.849 | 0.226 | 1.617 | 6.479 |
| **ResNet50/CIFAR-100** | Cosine | 0.013 | 5.186 | 1.187 | 0.009 | 1.295 | 1.256 | 0.056 | 2.450 | 1.433 | 0.022 | 3.599 | 1.206 |
| | Euclidean | 0.013 | 5.328 | 1.143 | 0.009 | 1.325 | 1.229 | 0.049 | 2.538 | 1.363 | 0.021 | 3.683 | 1.166 |
| | FedBary | 0.013 | 5.090 | 1.205 | 0.009 | 1.280 | 1.280 | 0.052 | 2.420 | 1.450 | 0.022 | 3.540 | 1.220 |
| | *IOS* | 0.013 | 4.995 | 1.223 | 0.009 | 1.264 | 1.308 | 0.053 | 2.399 | 1.474 | 0.023 | 3.491 | 1.238 |
| **BERT-base/20News** | Cosine | 0.259 | 1.145 | 5.980 | 0.404 | 0.909 | 10.612 | 0.281 | 0.970 | 6.795 | 0.246 | 1.378 | 6.331 |
| | Euclidean | 0.253 | 1.168 | 5.920 | 0.392 | 0.936 | 10.400 | 0.277 | 0.980 | 6.693 | 0.241 | 1.406 | 6.268 |
| | FedBary | 0.264 | 1.130 | 6.100 | 0.410 | 0.895 | 10.800 | 0.288 | 0.955 | 6.880 | 0.251 | 1.360 | 6.510 |
| | *IOS* | 0.268 | 1.118 | 6.214 | 0.416 | 0.884 | 10.980 | 0.292 | 0.944 | 6.984 | 0.256 | 1.346 | 6.582 |

Euclidean similarities between client gradient/importance vectors aggregated locally, and *FedBary*, a data-driven Wasserstein-based distance over client data distributions, chosen as a representative distribution-level metric whose similarity is computed from the overlap of clients' class-frequency histograms (ratio of records in the same classes). All baselines are *value-/gradient-based*, in contrast to the index-only nature of *IOS*. The results indicate that *IOS* consistently outperforms value-based methods across vision and text, with gains larger under stronger heterogeneity. On CIFAR-100 at $Dir(0.1)$, *IOS* attains $48.3\%$ vs. $45.40\%$ (Cosine), $43.18\%$ (Euclidean), and $46.00\%$ (*FedBary*).

*Cluster Quality Evaluation.* We assess clustering with Silhouette, Davies–Bouldin (DB), and Calinski–Harabasz (CH), where higher Sil/CH and lower DB are better. Using ground-truth label affinities, the *Oracle* is an upper bound. Table 2 shows that *IOS* reliably closes most of the gap to Oracle: per cell it is only ~15–20% lower on Sil/CH and ~14–20% higher on DB. Overall, *IOS* captures client relatedness more consistently than value-based similarities, yielding tighter intra-cluster cohesion and clearer inter-cluster separation.

**Neighbor Selection: Retrieval Quality vs. Oracle.** Personalized FL hinges on selecting the *right* peers: if the neighborhood assembled for a client does not mirror its underlying data distribution, no aggregation rule can reliably personalize downstream models. $\text{Recall}@k = \frac{1}{n}\sum_{i=1}^{n} \frac{|N_k^S(i) \cap N_k^\star(i)|}{k}$ directly measures how many oracle neighbors are recovered, while its trend over $k$ and across heterogeneity regimes reveals robustness. For each client $i$, let $N_k^\star(i)$ be the *oracle* neighbor set: the $k$ clients with the smallest Wasserstein distance between the *true* per-client label histograms (unavailable in practice; used only for evaluation). A method $S$ (Cosine, Euclidean, or *IOS*) returns $N_k^S(i)$ via $k$-NN on its similarity. In this experiment, $N = 40$, and we sweep $k \in \{4, 8, 16\}$.

Table 3 shows that *IOS* retrieves oracle neighbors more reliably than baselines, with larger gaps at higher $k$ and stronger non-IID. In the hardest regime (CIFAR-100, Dir(0.1)), *IOS* reaches R@8 $= 0.67$ vs. $0.61/0.53$ (Cosine/Euclidean) and $0.81$ at $k$=16 vs. $0.77/0.70$. Recall rises with $k$ for all methods, yet *IOS* keeps a lead (FMNIST, Dir(0.3), $k$=16: $0.97$ vs. $0.96/0.90$). As heterogeneity strengthens (Dir $0.3 \to 0.1$), all recalls drop but *IOS* degrades less (CIFAR-100 R@8: $0.70 \to 0.67$ vs. $0.66 \to 0.61$). The pattern is modality-agnostic: *IOS* tracks oracle structure in text (20News R@16 up to $0.97$), while vision under severe non-IID remains tough yet still favors *IOS*.

**Shapley-Style Donor Ranking.** In many PFL/DFL schemes, a client aggregates only from a few high-value *donors*. Inspired by Shapley-value notions of contributor utility in ML (Ghorbani & Zou, 2019; Wang et al., 2020; Lin et al., 2022), we *implement* a Shapley-style donor-ranking evaluation in our codebase: the *oracle* similarity $S^\star(i,j)$ via wasserstein distance over true label histograms induces the ground-truth donor order $\pi_\star(i)$. For any operational similarity $S$, we get a method order $\pi_S(i)$ by sorting row $S[i,\cdot]$ (excluding $i$). In our implementation, we replace Cosine with IOS: the default donor ranking uses *IOS* (index-overlap on Top-$K$ supports) rather than Cosine, and we report agreement with $\pi_\star(i)$ via Kendall's $\tau$ and $\text{Recall}@k$ (averaged over clients and final rounds).

Table 3: **Recall@k vs. Oracle** for neighbor selection. Within each regime, we report Recall@$k$.

| Dataset / Model | Method | $Dir(0.3)$ (k=4 / 8 / 16) | $Dir(0.1)$ (k=4 / 8 / 16) | $Patho(20\%)$ (k=4 / 8 / 16) | $Patho(30\%)$ (k=4 / 8 / 16) |
|---|---|---|---|---|---|
| CNN / FMNIST | Cosine | 0.64 / 0.81 / 0.96 | 0.58 / 0.75 / 0.90 | 0.55 / 0.72 / 0.88 | 0.56 / 0.73 / 0.89 |
| | Euclidean | 0.56 / 0.74 / 0.90 | 0.50 / 0.68 / 0.84 | 0.48 / 0.65 / 0.81 | 0.49 / 0.66 / 0.82 |
| | **IOS** | **0.70 / 0.88 / 0.97** | **0.65 / 0.78 / 0.92** | **0.58 / 0.74 / 0.90** | **0.59 / 0.78 / 0.94** |
| ResNet18 / CIFAR-10 | Cosine | 0.60 / 0.78 / 0.94 | 0.52 / 0.68 / 0.83 | 0.50 / 0.66 / 0.81 | 0.51 / 0.67 / 0.82 |
| | Euclidean | 0.53 / 0.72 / 0.89 | 0.44 / 0.60 / 0.77 | 0.42 / 0.58 / 0.75 | 0.43 / 0.59 / 0.76 |
| | **IOS** | **0.62 / 0.80 / 0.96** | **0.55 / 0.73 / 0.90** | **0.52 / 0.74 / 0.86** | **0.54 / 0.73 / 0.86** |
| ResNet50 / CIFAR-100 | Cosine | 0.48 / 0.66 / 0.82 | 0.44 / 0.61 / 0.77 | 0.42 / 0.59 / 0.75 | 0.43 / 0.60 / 0.76 |
| | Euclidean | 0.40 / 0.58 / 0.75 | 0.36 / 0.53 / 0.70 | 0.34 / 0.51 / 0.68 | 0.35 / 0.52 / 0.69 |
| | **IOS** | **0.49 / 0.70 / 0.85** | **0.45 / 0.67 / 0.81** | **0.44 / 0.63 / 0.79** | **0.45 / 0.64 / 0.78** |
| BERT-base / 20News | Cosine | **0.68 / 0.85 / 0.96** | 0.64 / 0.81 / 0.92 | 0.62 / 0.80 / 0.96 | 0.63 / 0.81 / 0.94 |
| | Euclidean | 0.61 / 0.79 / 0.95 | 0.57 / 0.75 / 0.92 | 0.55 / 0.73 / 0.90 | 0.56 / 0.74 / 0.91 |
| | **IOS** | 0.65 /0.82 / 0.97 | **0.64 / 0.83 / 0.94** | **0.63 / 0.82 / 0.95** | **0.65 / 0.83 / 0.97** |

Table 4: **Shapley-style donor ranking vs. oracle.** Each cell reports *Kendall's $\tau$ / Recall@5*. Methods: Cosine, Euclidean, and *IOS*. Oracle ranking is induced by $1-$JS on true label distributions.

| Dataset / Model | Method | $Dir(0.3)$ ($\tau$ / R@5) | $Dir(0.1)$ ($\tau$ / R@5) | $Patho(20\%)$ ($\tau$ / R@5) | $Patho(30\%)$ ($\tau$ / R@5) |
|---|---|---|---|---|---|
| CNN / FMNIST | Cosine | 0.48 / 0.63 | 0.41 / 0.57 | 0.39 / 0.55 | 0.40 / 0.56 |
| | Euclidean | 0.47 / 0.62 | 0.43 / 0.56 | 0.38 / 0.54 | 0.40 / 0.55 |
| | *IOS* | **0.59 / 0.68** | **0.52 / 0.62** | **0.49 / 0.60** | **0.51 / 0.61** |
| ResNet18 / CIFAR-10 | Cosine | 0.44 / 0.60 | 0.36 / 0.55 | 0.34 / 0.54 | 0.35 / 0.54 |
| | Euclidean | 0.43 / 0.59 | 0.35 / 0.53 | 0.34 / 0.53 | 0.34 / 0.53 |
| | **IOS** | **0.56 / 0.66** | **0.48 / 0.62** | **0.45 / 0.60** | **0.46 / 0.61** |
| ResNet50 / CIFAR-100 | Cosine | 0.32 / 0.52 | 0.28 / 0.49 | 0.27 / 0.48 | 0.28 / 0.49 |
| | Euclidean | 0.31 / 0.51 | 0.28 / 0.48 | 0.26 / 0.47 | 0.28 / 0.48 |
| | **IOS** | **0.41 / 0.56** | **0.36 / 0.52** | **0.34 / 0.50** | **0.35 / 0.51** |
| BERT-base / 20News | Cosine | 0.55 / 0.68 | 0.50 / 0.64 | 0.48 / 0.63 | 0.49 / 0.64 |
| | Euclidean | 0.54 / 0.67 | 0.49 / 0.62 | 0.47 / 0.62 | 0.48 / 0.62 |
| | **IOS** | **0.59 / 0.72** | **0.53 / 0.69** | **0.51 / 0.67** | **0.55 / 0.68** |

Table 4 shows *IOS* yields the highest agreement with the oracle, outperforming both Cosine and Euclidean. Gains are consistent across regimes: on CIFAR-10 with $Dir(0.1)$, *IOS* reaches $\tau=0.48$ / R@5 $= 0.62$ vs. $0.36/0.55$ (Cosine) and $0.35/0.53$ (Euclidean). On the harder CIFAR-100, $Dir(0.1)$, *IOS* attains $0.36/0.52$ vs. $0.28/0.49$ and $0.28/0.48$. For FMNIST, $Dir(0.3)$, *IOS* $0.59/0.68$ exceeds $0.48/0.63$ and $0.47/0.62$; for 20News, $Dir(0.3)$, *IOS* $0.59/0.72$ improves over $0.55/0.68$ and $0.54/0.67$. Euclidean trails Cosine by $\approx 0.01$–$0.02$ in both $\tau$ and R@5 throughout, reinforcing that *indices-only IOS* better preserves oracle donor priority without real-valued sharing.

**Oracle Distribution Alignment.** Personalization quality depends on whether a client's neighbor mixture reproduces its *true* data distribution. For client $i$ with label histogram $p_i$, and a method $S$ that selects a $k$-NN set $N_k^S(i)$ with weights $w_{ij} \propto S(i,j)$ (row-normalized), we form the induced mixture $\hat{p}_i^S(k) = \sum_{j \in N_k^S(i)} w_{ij}\, p_j$. We evaluate *alignment* via Jensen–Shannon divergence $JS\big(p_i, \hat{p}_i^S(k)\big)$ (lower is better), averaged over clients and the last 10 rounds. As an *oracle* upper bound, we compute $N_k^\star(i)$ using the $k$ neighbors that minimize the Wasserstein distance to $p_i$ (unavailable in practice; used only for evaluation). The result in Table 5 for fix $k=8$ indicates that *(i) IOS consistently yields tighter alignment to client distributions.* Across all settings, *IOS* is the best non-oracle: e.g., on *CIFAR-10* and $Dir(0.1)$, *IOS* reduces JS to 0.219 vs. 0.238 (Cosine, ↓0.019) and 0.252 (Euclidean, ↓0.033), approaching the oracle's 0.195 within 0.024. On the harder *CIFAR-100*, $Dir(0.1)$, *IOS* attains 0.285, improving over Cosine by 0.027 and over Euclidean by 0.044, and within 0.020 of oracle. *(ii) Non-IID severity increases mismatch for all, but* IOS *degrades least.* Moving from $Dir(0.3)$ to $Dir(0.1)$ increases JS by $\sim$0.02–0.04; IOS's increments are systematically smaller than

Table 5: **Oracle distribution alignment** (JS divergence; lower is better) with $k=8$ neighbors. Bold indicates the best *non-oracle* method.

| Dataset / Model | Dir(0.3) | | | | Dir(0.1) | | | | $Patho(20\%)$ | | | | $Patho(30\%)$ | | | |
|---|---|---|---|---|---|---|---|---|---|---|---|---|---|---|---|---|
| | Oracle | Cosine | Euclidean | **IOS** | Oracle | Cosine | Euclidean | **IOS** | Oracle | Cosine | Euclidean | **IOS** | Oracle | Cosine | Euclidean | **IOS** |
| **CNN / FMNIST** | 0.184 | 0.214 | 0.233 | **0.196** | 0.205 | 0.247 | 0.265 | **0.225** | 0.217 | 0.261 | 0.279 | **0.240** | 0.198 | 0.242 | 0.257 | **0.226** |
| **ResNet18 / CIFAR-10** | 0.173 | 0.205 | 0.226 | **0.188** | 0.195 | 0.238 | 0.252 | **0.219** | 0.208 | 0.259 | 0.275 | **0.237** | 0.189 | 0.236 | 0.249 | **0.221** |
| **ResNet50 / CIFAR-100** | 0.241 | 0.278 | 0.301 | **0.254** | 0.265 | 0.312 | 0.329 | **0.285** | 0.283 | 0.333 | 0.352 | **0.307** | 0.259 | 0.305 | 0.323 | **0.281** |
| **BERT-base / 20News** | 0.123 | 0.141 | 0.156 | **0.134** | 0.136 | 0.153 | 0.166 | **0.145** | 0.145 | 0.163 | 0.174 | **0.155** | 0.132 | 0.156 | 0.168 | **0.148** |

Cosine/Euclidean (e.g., *FMNIST*: +0.029 for *IOS* vs. +0.033 / +0.032). *(iii) Modality trend holds.* Text classification (*BERT-base/20News*) exhibits lower JS overall, yet *IOS* preserves a clear gap ($Dir(0.3)$: 0.134 vs. 0.141 / 0.156), showing that indices-only geometry transfers beyond vision.

## 6.3 SECURITY ANALYSIS

Two widely studied privacy threats in FL are reconstruction and membership inference attacks. Reconstruction attacks recover training examples by matching observed real-valued gradients, as in gradient inversion methods such as Zhu et al. (2019). Under IOS, the server never observes real-valued gradients, so per-parameter magnitudes required by gradient-matching objectives are absent, and known reconstruction techniques no longer apply in their standard form. In the remainder of this section, we therefore focus on membership inference attacks under the IOS information model.

**Membership Inference Attack (MIA)**. We evaluate a membership inference attack (MIA) adapted to IOS. In the baseline, the adversary observes full per-round raw gradients from the target client, enabling direct exploitation of gradient patterns for membership prediction. Under IOS, the adversary instead observes only the Top-$K$ index sets $I_i^t$ of salient coordinates per round $t$, computed from diagonal Fisher information, with no access to gradient values. For a target client $i$, the adversary aggregates $\{I_i^t\}_t$ across rounds and constructs a binary vector $b_i \in \{0, 1\}^M$, where $M$ is the number of model parameters and $b_i[m] = 1$ iff coordinate $m$ appears in the union $\bigcup_t I_i^t$. For a candidate data point $x$, the adversary trains a shadow model with the same architecture, computes its Fisher-based Top-$K$ salient index set, and encodes it as $b(x) \in \{0, 1\}^M$. Membership scores are computed as the index-overlap similarity between $b(x)$ and $b_i$ and compared against a fixed threshold. On a balanced member/non-member test set (random guessing $= 50\%$), gradient-based MIA using raw updates attains accuracies of 0.95/0.84/0.72/0.97 on CNN/ResNet18/ResNet50/BERT-base (avg. $\approx 87\%$), whereas the IOS-based variant drops to 0.71/0.49/0.56/0.69 (avg. 65.25%, Figure 1), only moderately above chance and thus substantially reducing—though not eliminating—membership leakage.

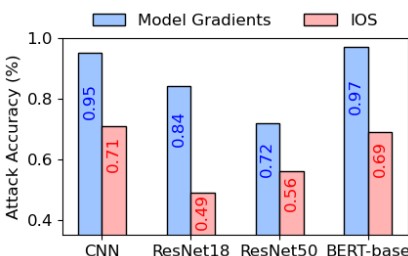

Figure 1: Attack success rates of gradient-based membership versus IOS.

## 7 DISCUSSION & CONCLUSION.

*IOS* is a value-free, index-only similarity —a drop-in replacement for value-based (parameter/gradient) affinities when sharing real numbers is undesirable. While our evaluation focuses on FL, the abstraction extends beyond federation: *IOS* applies to any domain that needs model-to-model similarity (e.g., model hubs, ensemble selection, continual/transfer learning, checkpoint curation) where value sharing is costly or risky. By measuring *support overlap* of salient parameters, it captures the inductive bias most predictive of relatedness, yielding consistent gains across applications: clustered PFL (faster convergence, higher accuracy), neighbor selection (higher Recall@$k$), donor ranking (higher Kendall-$\tau$), and distribution alignment (lower JS). Exchanging indices only *shrinks the attack surface* (gradient leakage, model inversion, membership/property inference). Overall, *IOS* recovers near-oracle structure with lightweight communication and a stronger privacy posture—advancing scalable personalization under communication and privacy constraints.

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

## A   HASH–LSH FOR SCALABLE CANDIDATE RETRIEVAL

**Why Hash–LSH.**   Computing all-pairs overlap between $n$ clients' supports $\{I_i\}_{i=1}^n$ costs $O(n^2 K)$ even with sorted lists/bitsets. We use MinHash with locality-sensitive hashing (LSH) to retrieve *candidate neighbors* in subquadratic time and bytes: each client publishes a short signature (or only band buckets), the server (or a peer) looks up candidates that collide in at least one band, and exact similarity is computed only on this small candidate set. This preserves *IOS*'s value-free property (only indices/signatures, no real-valued parameters) and scales to large $n$. For a support set $I \subseteq [M]$, define $h$ independent minhashes $\phi_k(I) = \min\{\, H_k(j) : j \in I \,\}$ using 2-universal (approx. minwise) hash functions $H_k : [M] \rightarrow \{0, \ldots, 2^{64} - 1\}$. The signature is $\Phi(I) = (\phi_1(I), \ldots, \phi_h(I)) \in \mathbb{N}^h$. For any two sets $I_i, I_j$ with $S(i,j)$, an an unbiased estimator with $\mathrm{Var}[\hat{s}] = s(1-s)/h$ is $\mathbb{P}\big[\, \phi_k(I_i) = \phi_k(I_j)\,\big] = s \quad \Rightarrow \quad \hat{s} = \frac{1}{h}\sum_{k=1}^h \mathbf{1}\{\phi_k(I_i) = \phi_k(I_j)\}$. Signature cost is $O(hK)$ time and $O(h)$ words per client.

---

**Algorithm 2** IOS–MinHash–LSH (build & query)

---

**Input:** Supports $\{I_i\}_{i=1}^n$, hash family $\{H_k\}_{k=1}^h$, bands $b$, rows per band $r$, exact index store for $I_i$

1: **Build:** For each client $i$:
2:     Compute signature $\Phi(I_i)$ where $\phi_k(I_i) = \min_{j \in I_i} H_k(j)$
3:     For each band $u = 1..b$, form key $B_u(I_i) = (\phi_{(u-1)r+1}, \ldots, \phi_{ur})$ and insert $i$ into table $\mathcal{T}_u[B_u(I_i)]$
4: **Query($q$):** Given a query support $I_q$
5:     Compute $\Phi(I_q)$ and band keys $B_u(I_q)$
6:     Candidates $C \leftarrow \bigcup_{u=1}^b \mathcal{T}_u[B_u(I_q)] \setminus \{q\}$
7:     For each $j \in C$: fetch $I_j$ (if not local), compute $S(q,j) = \frac{|I_q \cap I_j|}{K}$
8:     Return top-$k$ by $S(q,j)$ (or all with $S(q,j) \geq s_{\min}$)

---

## B  ALIGNMENT PROOFS (OVERLAP-ONLY)

### B.1  AUXILIARY BOUND

**Lemma 1** (Per-coordinate lower bound from dispersion). *If $\kappa_i \leq \kappa$, then for every $u \in I_i$,*

$$(h_i)_u \; \geq \; \frac{\|h_i\|_2}{\kappa\sqrt{K}}.$$

*Proof.* Let $a = \min_{u \in I_i}(h_i)_u$ and $b = \max_{u \in I_i}(h_i)_u \leq \kappa a$. Then $\|h_i\|_2^2 = \sum_{u \in I_i}(h_i)_u^2 \leq Kb^2 \leq K\kappa^2 a^2$, so $a \geq \|h_i\|_2/(\kappa\sqrt{K})$. □

### B.2  PROOF OF PROPOSITION 1

Because $s_i, s_j$ are nonnegative,

$$\langle s_i, s_j \rangle \; \geq \; \sum_{u \in I_i \cap I_j}(h_i)_u(h_j)_u \; \geq \; |I_i \cap I_j| \cdot \frac{\|h_i\|_2}{\kappa\sqrt{K}} \cdot \frac{\|h_j\|_2}{\kappa\sqrt{K}} \; = \; \frac{|I_i \cap I_j|}{\kappa^2 K}\|h_i\|_2\|h_j\|_2 \; = \; \frac{R_{ij}}{\kappa^2}\|h_i\|_2\|h_j\|_2.$$

using Lemma 1. Head dominance gives $\|h_i\|_2 \geq \sqrt{1-\varepsilon}\,\|s_i\|_2$ and similarly for $j$. Divide by $\|s_i\|_2\|s_j\|_2$ to obtain equation 2. □

## C  STABILITY PROOFS (EMA + UNION BOUND)

### C.1  EMA VARIANCE

Let $\eta^{(t)} = \sum_{s \geq 0}(1-\beta)\beta^s \xi^{(t-s)}$ be the EMA noise at any coordinate (client index suppressed). Since the $\xi^{(t)}$ are independent, mean-zero, sub-Gaussian with proxy variance $\sigma^2$,

$$\mathrm{Var}(\eta^{(t)}) \; = \; \sigma^2 \sum_{s \geq 0}(1-\beta)^2\beta^{2s} \; = \; \sigma^2\frac{(1-\beta)^2}{1-\beta^2} \; = \; \sigma^2\frac{1-\beta}{1+\beta}.$$

### C.2  CONCENTRATION AT THE BOUNDARY

Let $H$ be the set of the $K$ largest means of $\mu$ and $\bar{H} = [M] \setminus H$. For $u \in H$,

$$\mathrm{Pr}\Big(\tilde{g}_u^{(t)} < \mu_{(K)} - \tfrac{\Delta}{2}\Big) \; \leq \; \exp\Big(-\tfrac{\Delta^2}{8\,\sigma(\beta)^2}\Big).$$

A union bound over the $K$ elements of $H$ yields $\mathrm{Pr}(\min_{u \in H}\tilde{g}_u^{(t)} < \mu_{(K)} - \Delta/2) \leq K\exp(-\Delta^2/(8\sigma(\beta)^2))$. Similarly, for $v \in \bar{H}$, $\mathrm{Pr}(\tilde{g}_v^{(t)} > \mu_{(K+1)} + \Delta/2) \leq \exp(-\Delta^2/(8\sigma(\beta)^2))$, and union over $M - K$ indices gives $\mathrm{Pr}(\max_{v \in \bar{H}}\tilde{g}_v^{(t)} > \mu_{(K+1)} + \Delta/2) \leq (M - K)\exp(-\Delta^2/(8\sigma(\beta)^2))$. Union over these two bad events proves Theorem 1. □

### C.3  EXPECTED TEMPORAL SELF-OVERLAP

Let $\mathcal{B}$ be the subset of $K$-boundary positions whose mean gaps exceed $c\,\sigma(\beta)\sqrt{\log M}$; assume $|\mathcal{B}|/K \geq \rho$. On the event guaranteed by Theorem 1 at times $t - 1$ and $t$, the indices in $\mathcal{B}$ persist in the Top-$K$. Taking expectation over the complement yields $\mathbb{E}[\Gamma^{(t)}] \geq \rho - O(M^{1-c^2/8})$. □

### C.4 EXTENSIONS BEYOND SUB-GAUSSIAN NOISE

**Remark C.4 (Beyond sub-Gaussian noise).** Theorem 1 is stated under independent, mean-zero sub-Gaussian coordinates for the base noise $\xi^{(t)}$ at each coordinate, but the argument in Appendix C only uses two ingredients: (i) an exponential tail bound at the coordinate level and (ii) independence across time to obtain the EMA variance factor $\sigma(\beta)^2 = \sigma^2 \frac{1-\beta}{1+\beta}$ derived in §C.1. The union bound over $M$ coordinates does not require independence across coordinates. Under sub-exponential noise, essentially the same concentration step goes through with a modified tail bound and a different dependence on the boundary margin $\Delta$, yielding a qualitatively identical "large margin $\Rightarrow$ high stability" conclusion as in Theorem 1. Truly heavy-tailed noise would require either stronger margin conditions or robustified importance estimators (e.g., truncation or median-of-means), which we view as promising future work rather than a prerequisite for the current theorem.

# D  MORE THEORETICAL ANALYSIS

## D.1  ROBUSTNESS TO RE-SCALING AND DIAGONAL PRECONDITIONING

**Setting (global Top-$K$).**  We select a single global set $(g, K) \subseteq [M]$ over the concatenated parameter vector; *no per-layer budgets* are used. The results below distinguish (i) invariances under positive scalings, (ii) effects of blockwise (layer-constant) rescaling, and (iii) diagonal preconditioners.

**Lemma C.1 (Positive scalar invariance; blockwise order preservation).**  Let importance be monotone in $|g|$ or $g^2$. Then, for any $a > 0$,

$$(ag, K) = (g, K).$$

Moreover, if $D$ rescales each layer $\ell$ by a positive constant $d_\ell > 0$, the *within-layer* ranking of coordinates is unchanged, although the *global* Top-$K$ membership may change via cross-layer swaps. *Proof.* Positive scalar scaling preserves all pairwise orders; blockwise scaling preserves orders within blocks (layers). □

**Lemma C.2 (Cross-layer stability under bounded block rescaling).**  Let $D = \mathrm{diag}(d_u)$ with $d_u = d_\ell$ for all $u$ in layer $\ell$, and block scales $d_\ell \in [1/\chi, \chi]$. Let $I^\star = (g, K)$ and suppose $g$ has dispersion margin $\kappa > 1$ at the boundary: for every pair $(u, v)$ with $u \in I^\star$, $v \notin I^\star$, we have $g_u/g_v \geq \kappa$. Then

$$\frac{\left| (Dg, K) \triangle I^\star \right|}{K} \leq \eta(\kappa, \chi, K),$$

where $\eta(\kappa, \chi, K)$ counts boundary pairs whose ratio lies in $[1, \chi^2)$ (i.e., near-ties that block rescaling can invert). In particular, $\eta(\kappa, \chi, K) \downarrow 0$ as $\chi \downarrow 1$ or as $\kappa$ increases. *Proof.* For any boundary pair $(u, v)$, the post-rescaling ratio is $(d_{\ell(u)}/d_{\ell(v)}) \cdot (g_u/g_v) \in [\kappa/\chi^2, \kappa\chi^2]$. If $\kappa > \chi^2$, the order is preserved. Violations can only arise from near-ties, which bounds the symmetric difference. □

**Corollary C.3 (Diagonal preconditioners).**  Let $D_t = \mathrm{diag}(d_{t,u})$ be a diagonal preconditioner (e.g., Adam's $\hat{v}_t^{-1/2}$) with overall condition number $\chi_t = \frac{\max_u d_{t,u}}{\min_u d_{t,u}} \leq \bar{\chi}$. Then, across a round,

$$\frac{\left| (D_t g, K) \triangle (g, K) \right|}{K} \leq \eta(\kappa, \bar{\chi}, K).$$

*Proof.* For any boundary pair, the scaled ratio lies in $[\kappa/\bar{\chi}^2, \kappa\bar{\chi}^2]$; the same near-tie argument as in Lemma C.2 applies. □

## D.2  CHOOSING $K$: COVERAGE, STABILITY, AND SAMPLE COMPLEXITY

**Proposition D.1 (Monotone coverage; selection complexity).**  Let $h_i$ denote the head of length $K$ after ordering coordinates of $g_i$ by decreasing importance. Then $C_i(K) = \|h_i\|_2^2 / \|g_i\|_2^2$ is non-decreasing in $K$. The smallest $K \leq K_{\max}$ with $C_i(K) \geq \tau$ can be found by binary search in $O(\log K_{\max})$ iterations, each using a selection step that is $O(M)$ expected time (via Quickselect) or $O(M \log M)$ with sorting. *Proof.* Adding indices cannot reduce head energy; complexity follows from selection/sorting costs. □

**Proposition D.2 (Stability estimator concentration).**  Let $\hat{\rho}_i(K) = \frac{1}{r} \sum_{s=1}^{r} Z_s$ where $Z_s \in \{0, 1\}$ indicates whether the Top-$K$ set is unchanged between two adjacent EMA slices (or light bootstrap resamples). Then, for any $\delta \in (0, 1)$,

$$\Pr\left( |\hat{\rho}_i(K) - \rho_i(K)| > \delta \right) \leq 2 \exp\left( -2r\delta^2 \right).$$

Thus $r \geq \frac{1}{2\delta^2} \log \frac{2}{\gamma}$ samples suffice for accuracy $\delta$ with confidence $1 - \gamma$. *Proof.* Hoeffding's inequality. □

**Remark D.3 (End-to-end frontier).**  Choosing $(\tau, \rho_0, r, \beta)$ traces a utility–privacy–bandwidth curve. Larger $\beta$ improves $\hat{\rho}$ but slows drift response; $r$ trades runtime for confidence.

### D.3 MinHash–LSH facts and end-to-end complexity

**Lemma E.1 (MinHash unbiasedness and variance).** Let $S_1, S_2 \subseteq [M]$ and $s = J(S_1, S_2) = \frac{|S_1 \cap S_2|}{|S_1 \cup S_2|}$. With $h$ independent MinHash functions, the estimator $\hat{s} = \frac{1}{h} \sum_{u=1}^{h} \mathbf{1}\{\mathrm{mh}_u(S_1) = \mathrm{mh}_u(S_2)\}$ satisfies

$$\mathbb{E}[\hat{s}] = s, \qquad \mathrm{Var}[\hat{s}] = \frac{s(1-s)}{h}.$$

*Proof.* Each indicator is Bernoulli($s$) by MinHash collision equivalence. □

**Lemma E.2 (Banding retrieval probability).** Arrange $h = br$ MinHash values into $b$ bands of $r$ rows; two sets with similarity $s$ collide in a band with probability $s^r$, and are retrieved with probability $1 - (1 - s^r)^b$. *Proof.* Standard LSH banding analysis with independent bands. □

**Theorem E.3 (Candidate set size and total cost).** Let $n$ clients each submit a $K$-subset (the TopK indices). Using $h = br$ MinHashes and $b$ hash tables: expected candidate set size per query is

$$\mathbb{E}[C] = 1 + \sum_{q \neq i} \left( 1 - (1 - s_{iq}^r)^b \right),$$

where $s_{iq}$ are pairwise similarities. The total server cost per round is

$$O(nh) + O\left( \sum_{i=1}^{n} \mathbb{E}[C_i] \right) + O\left( \sum_{(i,q) \in \mathrm{cands}} |I_i \cap I_q| \right),$$

i.e., signature build + candidate lookups + final exact overlaps, with the last term bounded by $O(\sum_i \mathbb{E}[C_i] \cdot K)$. Signature memory is $O(nh)$. *Proof.* Linearity of expectation and that exact overlap is $O(\min(K_i, K_q)) = O(K)$. □

### D.4 Computation and Communication Analysis ($K \ll M$)

**Computational overhead.** Let $C_{\mathrm{train}} = S \cdot C_{\mathrm{fb}}(M)$ denote the per-round local training cost for $S$ SGD steps on a model with $M$ parameters, where $C_{\mathrm{fb}}(M)$ is the cost of one forward–backward pass. IOS adds two $O(M)$ components on the client: (i) an EMA update of per-parameter squared gradients at each step, with per-step cost $C_{\mathrm{ema}}(M)$, and (ii) a once-per-round Top-$K$ selection on $\mathbb{R}^M$ with cost $C_{\mathrm{topK}}(M, K)$ (e.g., selection + partial sort). The additional per-round cost is

$$C_{\mathrm{IOS}} = S \cdot C_{\mathrm{ema}}(M) + C_{\mathrm{topK}}(M, K),$$

and the relative overhead is

$$\frac{C_{\mathrm{IOS}}}{C_{\mathrm{train}}} = \frac{C_{\mathrm{ema}}(M)}{C_{\mathrm{fb}}(M)} + \frac{C_{\mathrm{topK}}(M, K)}{S \cdot C_{\mathrm{fb}}(M)}.$$

Since $C_{\mathrm{fb}}(M)$ is dominated by convolutions and matrix multiplies, while $C_{\mathrm{ema}}(M)$ and $C_{\mathrm{topK}}(M, K)$ are simple linear-time vector operations, both terms are small for typical FL settings where $S$ is at least on the order of tens of steps.

**Communication overhead.** When full models are already uploaded for aggregation (e.g., CPFL), a 32-bit model costs $B_{\mathrm{model}} = 32M$ bits. A Top-$K$ index set uses

$$B_{\mathrm{idx}} = K \cdot \lceil \log_2 M \rceil \approx 32K$$

bits if indices are stored as 32-bit integers, giving a relative overhead

$$\frac{B_{\mathrm{idx}}}{B_{\mathrm{model}}} \approx \frac{K}{M} = \rho.$$

In similarity-only tasks where a standard design would upload either the full parameter vector ($B_{\mathrm{model}} = 32M$ bits) or a dense embedding of size $D$ ($B_{\mathrm{embed}} = 32D$ bits), IOS instead sends either the raw index set ($B_{\mathrm{idx}}$ above) or a MinHash signature of length $L$ with $b_{\mathrm{hash}}$-bit hashes, costing $B_{\mathrm{hash}} = L \cdot b_{\mathrm{hash}}$ bits. The resulting compression factors relative to full models are

$$\frac{B_{\mathrm{model}}}{B_{\mathrm{idx}}} = \Theta\left( \frac{M}{K} \right) = \Theta\left( \frac{1}{\rho} \right), \qquad \frac{B_{\mathrm{model}}}{B_{\mathrm{hash}}} = \Theta\left( \frac{M}{L} \right).$$

For instance, with $M \approx 2.5 \times 10^7$, $K = 0.15M$, $L = 256$, and $b_{\text{hash}} = 32$, we obtain $\frac{B_{\text{model}}}{B_{\text{idx}}} \approx 6.7\times$ and $\frac{B_{\text{model}}}{B_{\text{hash}}} \approx 10^5\times$ compression, illustrating that IOS yields several-fold savings with raw index sets and up to two orders of magnitude with short MinHash signatures in the regimes $K = \rho M$ with $\rho < 1$ and $L \ll M$.

Table 6: Empirical computation time on an A100 for baseline training vs. IOS (Fisher + Top-$K$) instrumentation. IOS overhead is split into importance analysis (Fisher accumulation) and a once-per-round Top-$K$ selection; their sum is reported along with its percentage of baseline training time.

| Model / Dataset | $K/M$ (%) | Training (s) | Importance analysis (s) | Top-$K$ selection (s) | IOS overhead (s) [%] |
|---|---|---|---|---|---|
| CNN / FMNIST | 12.2 | 4.00 | 0.14 | 0.02 | 0.16 (4.0%) |
| ResNet18 / CIFAR-10 | 7.3 | 6.50 | 0.15 | 0.05 | 0.20 (3.1%) |
| ResNet50 / CIFAR-100 | 9.6 | 13.00 | 0.28 | 0.11 | 0.39 (3.0%) |
| BERT-base / 20News | 8.2 | 18.00 | 0.28 | 0.17 | 0.45 (2.5%) |

# E    EXTENDED EXPERIMENTAL RESULTS

## E.1    EMPIRICAL COMPUTATIONAL AND COMMUNICATION OVERHEAD

To complement the complexity analysis in §4.4, we empirically estimate the wall-clock cost of IOS on the four model/dataset pairs used throughout the paper: a 2-layer CNN on FMNIST, ResNet18 on CIFAR-10, ResNet50 on CIFAR-100, and BERT-base on 20News. For each model we measure: (i) the time to run 5 local training epochs with standard SGD; and (ii) the additional time to maintain the EMA Fisher proxy and extract a Top-$K$ index set once per round (IOS instrumentation), using the same support sizes as in our main experiments ($K/M = 12.2\%, 7.3\%, 9.6\%, 8.2\%$ for CNN, ResNet18, ResNet50, and BERT-base, respectively). Updating the EMA of squared gradients requires a single fused multiply–add per parameter (identical to one Adam moment update). On GPU this is a single vectorized kernel and adds ¡0.1 ms per batch, which is below the timing resolution of our profiler. As a result, EMA contributes negligible runtime overhead compared to the backward pass and we omit it from Table X, which reports only the measurable Top-K cost. All measurements use a single NVIDIA A100 (40 GB) and a 32-core CPU, matching our main experimental setup.

**Computation.**    Table 6 breaks down the computation cost of a training phase into baseline training and the two IOS components: importance analysis (Fisher accumulation) and a once-per-round Top-$K$ selection. Importance analysis is the dominant term, adding only 0.14–0.28 s per training phase across models, while the Top-$K$ step is even smaller, ranging from 0.02 s on the CNN up to 0.17 s on BERT-base. Summed together, IOS adds just 0.16–0.45 s of extra time per training phase: about 4% for the small CNN and between 2.5% and 3.1% for ResNet18, ResNet50, and BERT-base.    Concretely, for ResNet50 on CIFAR-100, baseline training takes 13.00 s, and IOS adds only 0.28 s for importance analysis and 0.11 s for Top-$K$, for a total of 0.39 s (roughly 3% overhead). For BERT-base on 20News, the absolute IOS cost is still below half a second (0.28 s + 0.17 s), which corresponds to only 2.5% of the training phase. These examples illustrate that IOS instrumentation induces only a tiny constant-factor overhead even on deeper convolutional and transformer architectures.

**Communication.**    We also translate the bit complexity into concrete uplink times at a nominal 500 Mb/s link. Sending a full 32-bit model with $M$ parameters costs $B_{\text{model}} = 32M$ bits, while sending only the Top-$K$ support indices costs $B_{\text{idx}} = 32K$ bits (using 32-bit integer indices). Table 7 reports the corresponding one-shot upload times for the four models using the same $K/M$ ratios as above. Even at a relatively fast link, full-model payloads already cost $\approx$ 0.7–1.6 s per round for ResNet18/50 and about 7 s for BERT-base, whereas IOS reduces this to tens or hundreds of milliseconds (8–14$\times$ compression for the $K/M$ used in our experiments). When models must be sent anyway (e.g., for aggregation), IOS adds a negligible extra $K/M$ fraction; when similarity alone is needed, IOS replaces tens–hundreds of MB of real-valued state by a small integer sketch.

## E.2    EFFECT OF THE SUPPORT SIZE $K$ FOR *IOS*

For each model/dataset we compare three *IOS* support sizes: the task-specific $K^\star$ (selected by our coverage+stability rule; typically $\approx$7–12% and concretely 12.2% for CNN/FMNIST, 7.3% for ResNet18/CIFAR-10, 9.6% for ResNet50/CIFAR-100, and 8.2% for BERT-base/20News), plus two

Table 7: One-shot uplink time at 500 Mb/s for uploading a full 32-bit model vs. the IOS Top-$K$ index set ($K/M$ as in our experiments). "Reduction" is the factor by which IOS reduces the payload.

| Model / Dataset | $M$ (params) | Model upload (s) | IOS indices (s) | Reduction ($\times$) |
|---|---|---|---|---|
| CNN / FMNIST | $1.0 \times 10^6$ | 0.064 | 0.008 | $8.2\times$ |
| ResNet18 / CIFAR-10 | $1.1 \times 10^7$ | 0.704 | 0.051 | $13.7\times$ |
| ResNet50 / CIFAR-100 | $2.5 \times 10^7$ | 1.600 | 0.154 | $10.4\times$ |
| BERT-base / 20News | $1.1 \times 10^8$ | 7.040 | 0.577 | $12.2\times$ |

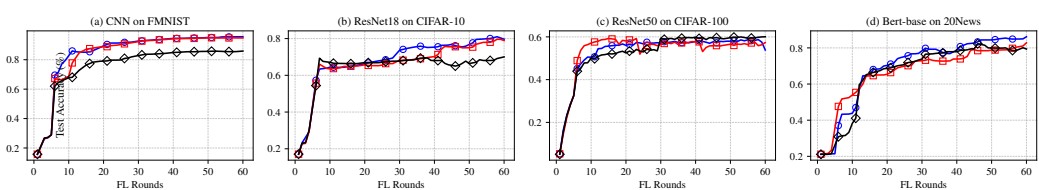

Figure 2: Comparative test accuracy of model based on FL rounds ($Patho(20\%)$).

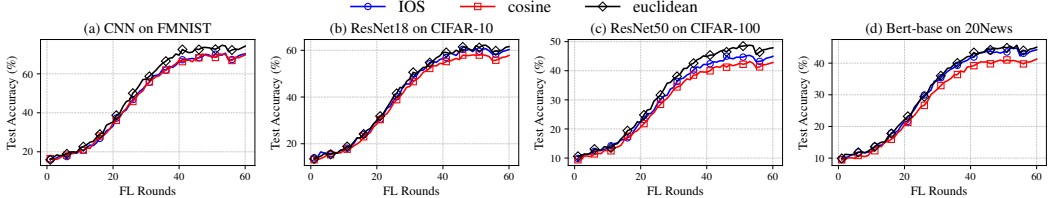

Figure 3: Comparative test accuracy of model based on FL rounds ($Dir(0.1)$).

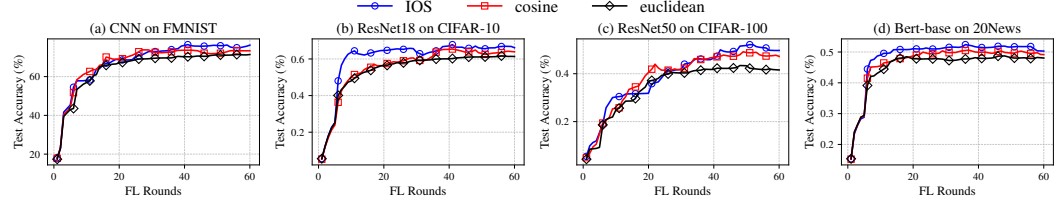

Figure 4: Comparative test accuracy of model based on FL rounds ($Dir(0.3)$).

larger supports 15% and 25%. All other settings (non-IID regimes, data splits, training budgets, and pipelines) match §6.

**Clustered Personalized FL (CPFL) accuracy vs. $K$.** We run IFCA/Clustered-FL style training with the *same* clustering/training pipeline while varying only $K$ in *IOS* to quantify how support size impacts downstream CPFL test accuracy.

Table 8 reports CPFL accuracy (%) across four heterogeneity regimes; within each block, columns are $K^\star$, 15%, and 25%.

$K^\star$ is best across all models/regimes. Moving to 15% costs roughly 2–6 pp depending on task/heterogeneity, while 25% introduces larger losses (typically 4–10 pp, up to ∼10 pp on the hardest cases). Deeper models under stronger heterogeneity suffer more. On *ResNet50/CIFAR-100* under Dir(0.1), accuracy falls from 48.30 at $K^\star$ to 44.20 at 15% ($-4.10$ pp) and 38.50 at 25% ($-9.80$ pp). For *BERT-base/20News* under Patho(20%), the score decreases from 58.26 to 55.86 at 15% ($-2.40$ pp) and to 51.96 at 25% ($-6.30$ pp). On the lighter *CNN/FMNIST* regime Dir(0.3), accuracy drops from 76.23 to 73.23 at 15% ($-3.00$ pp) and to 70.03 at 25% ($-6.20$ pp). The results

Table 8: **CPFL Accuracy (%) with *IOS*** at three $K$ settings. Within each distribution block, columns are $K^\star$, 15%, and 25%. Bold marks the best within each block.

| Model / Dataset | $Dir(0.3)$ | | | $Dir(0.1)$ | | | $Patho(20\%)$ | | | $Patho(30\%)$ | | |
| --- | --- | --- | --- | --- | --- | --- | --- | --- | --- | --- | --- | --- |
| | $K^\star$ | 15% | 25% | $K^\star$ | 15% | 25% | $K^\star$ | 15% | 25% | $K^\star$ | 15% | 25% |
| **CNN / FMNIST** | **76.23** | 73.23 | 70.03 | **74.79** | 71.29 | 67.39 | **95.18** | 92.98 | 90.38 | **83.15** | 80.45 | 77.65 |
| **ResNet18 / CIFAR-10** | **66.13** | 63.53 | 60.13 | **62.10** | 58.70 | 54.90 | **81.44** | 78.54 | 74.84 | **74.81** | 72.41 | 69.61 |
| **ResNet50 / CIFAR-100** | **49.71** | 45.81 | 40.71 | **48.30** | 44.20 | 38.50 | **58.07** | 54.87 | 49.77 | **52.98** | 49.48 | 44.08 |
| **BERT-base / 20News** | **50.30** | 48.10 | 45.60 | **45.41** | 43.41 | 40.31 | **58.26** | 55.86 | 51.96 | **54.25** | 52.15 | 49.05 |

Table 9: **Neighbor Selection Quality** (*Recall@8* vs. oracle) with *IOS* at three $K$ settings. Higher is better. Bold marks the best within each block.

| Model / Dataset | $Dir(0.3)$ | | | $Dir(0.1)$ | | | $Patho(20\%)$ | | | $Patho(30\%)$ | | |
| --- | --- | --- | --- | --- | --- | --- | --- | --- | --- | --- | --- | --- |
| | $K^\star$ | 15% | 25% | $K^\star$ | 15% | 25% | $K^\star$ | 15% | 25% | $K^\star$ | 15% | 25% |
| **CNN / FMNIST** | **0.88** | 0.85 | 0.82 | **0.78** | 0.74 | 0.70 | **0.74** | 0.71 | 0.67 | **0.78** | 0.75 | 0.71 |
| **ResNet18 / CIFAR-10** | **0.80** | 0.77 | 0.73 | **0.73** | 0.69 | 0.65 | **0.74** | 0.70 | 0.65 | **0.73** | 0.70 | 0.66 |
| **ResNet50 / CIFAR-100** | **0.70** | 0.66 | 0.61 | **0.67** | 0.63 | 0.57 | **0.63** | 0.60 | 0.55 | **0.64** | 0.61 | 0.56 |
| **BERT-base / 20News** | **0.82** | 0.80 | 0.77 | **0.83** | 0.81 | 0.77 | **0.82** | 0.80 | 0.77 | **0.83** | 0.81 | 0.78 |

shows that Small, stable supports near the coverage/stability knee ($K^\star$) give the best downstream accuracy; enlarging $K$ dilutes salience, destabilizes Top-$K$ boundaries, and consistently reduces CPFL performance, especially in harder regimes and deeper nets.

**Neighbor selection quality (Recall@8) vs. $K$.** We build the similarity graph with *IOS* at each $K$ and perform $k$-NN ($k=8$) neighbor selection. We then compute Recall@8 against the oracle neighbors (defined by Wasserstein distance over true per-client label histograms; used only for evaluation).

Table 9 reports Recall@8 (higher is better) across the four heterogeneity regimes; within each block, columns are $K^\star$, 15%, and 25%. The results indicate that $K^\star$ yields the highest Recall@8 throughout. Increasing to 15% reduces Recall by roughly 0.02–0.05; to 25% by about 0.05–0.09. Losses are more pronounced in harder regimes and for deeper models (e.g., ResNet50 under Dir(0.1)). For *ResNet50/CIFAR-100* under Dir(0.1), Recall@8 declines from 0.67 at $K^\star$ to 0.63 at 15% ($-0.04$) and 0.57 at 25% ($-0.10$). On *ResNet18/CIFAR-10* with Patho(20%), it goes from 0.74 to 0.70 at 15% ($-0.04$) and 0.65 at 25% ($-0.09$). For *BERT-base/20News* under Dir(0.3), Recall@8 moves from 0.82 to 0.80 at 15% ($-0.02$) and 0.77 at 25% ($-0.05$). Neighbor retrieval quality tracks the same pattern as downstream accuracy: supports near $K^\star$ best preserve *salient* head overlap across clients, while larger supports introduce tail coordinates that dilute overlap and make Top-$K$ less stable—lowering Recall and, downstream, CPFL performance.

### E.3    TRAINING EPOCHS ABLATION FOR *IOS*

**Protocol.** Unless stated otherwise, our study uses 8 local epochs. Here we ablate the number of local epochs $\{2, 4, 6, 8, 10\}$ while *fixing IOS* at the task-specific $K^\star$ (coverage+stability rule) and keeping all other settings (non-IID regimes, data splits, budgets, optimizers, and pipelines) identical to §6. The goal is to test how many epochs are sufficient for clients to reflect their underlying distributions in both downstream CPFL accuracy and neighbor retrieval.

**Clustered Personalized FL (CPFL) accuracy vs. epochs.** We run IFCA/Clustered-FL with the same clustering/training pipeline and vary only the number of local epochs. We report CPFL test accuracy (%) across all four heterogeneity regimes.

Table 10 shows CPFL accuracy for *IOS* at $K^\star$ under epochs $2, 4, 6, 8, 10$. The results show that accuracy increases steadily with more local training and saturates by 8 epochs: moving from $6 \to 8$ yields small gains (typically $+1$–$7$ pp over 6), and 10 epochs bring only marginal improvements

Table 10: **CPFL Accuracy (%) with *IOS* at $K^\star$** across local epochs. Within each distribution block, columns are **2 / 4 / 6 / 8 / 10** epochs. Bold marks the best within each block.

| Model / Dataset | Dir(0.3) | | | | | Dir(0.1) | | | | | Patho(20%) | | | | | Patho(30%) | | | | |
|---|---|---|---|---|---|---|---|---|---|---|---|---|---|---|---|---|---|---|---|---|
| | 2 | 4 | 6 | 8 | 10 | 2 | 4 | 6 | 8 | 10 | 2 | 4 | 6 | 8 | 10 | 2 | 4 | 6 | 8 | 10 |
| **CNN / FMNIST** | 63.90 | 69.70 | 72.60 | 76.23 | **76.45** | 62.50 | 66.90 | 71.10 | 74.79 | **75.05** | 83.40 | 86.70 | 92.20 | 95.18 | **95.30** | 72.00 | 76.10 | 80.60 | 83.15 | **83.45** |
| **ResNet18 / CIFAR-10** | 53.90 | 58.40 | 62.10 | 66.13 | **66.35** | 49.10 | 53.60 | 57.80 | 62.10 | **62.30** | 70.20 | 74.10 | 78.00 | 81.44 | **81.72** | 63.20 | 68.40 | 71.90 | 74.81 | **75.00** |
| **ResNet50 / CIFAR-100** | 36.00 | 40.90 | 44.30 | 49.71 | **49.95** | 35.10 | 39.90 | 43.40 | 48.30 | **48.50** | 46.50 | 50.20 | 54.40 | 58.07 | **58.25** | 40.90 | 45.90 | 49.70 | 52.98 | **53.12** |
| **BERT-base / 20News** | 39.00 | 43.90 | 47.50 | 50.30 | **50.48** | 33.70 | 38.90 | 42.50 | 45.41 | **45.60** | 47.90 | 52.30 | 55.80 | 58.26 | **58.38** | 44.50 | 48.90 | 52.10 | 54.25 | **54.34** |

Table 11: **Neighbor Selection Quality** (*Recall@8* vs. oracle) with *IOS* at $K^\star$ across epochs. Within each distribution block, columns are **2 / 4 / 6 / 8 / 10** epochs. Higher is better; bold marks the best.

| Model / Dataset | Dir(0.3) | | | | | Dir(0.1) | | | | | Patho(20%) | | | | | Patho(30%) | | | | |
|---|---|---|---|---|---|---|---|---|---|---|---|---|---|---|---|---|---|---|---|---|
| | 2 | 4 | 6 | 8 | 10 | 2 | 4 | 6 | 8 | 10 | 2 | 4 | 6 | 8 | 10 | 2 | 4 | 6 | 8 | 10 |
| **CNN / FMNIST** | 0.75 | 0.80 | 0.84 | 0.88 | **0.89** | 0.66 | 0.70 | 0.74 | 0.78 | **0.79** | 0.60 | 0.65 | 0.70 | 0.74 | **0.75** | 0.65 | 0.70 | 0.74 | 0.78 | **0.79** |
| **ResNet18 / CIFAR-10** | 0.67 | 0.72 | 0.76 | 0.80 | **0.81** | 0.58 | 0.64 | 0.69 | 0.73 | **0.74** | 0.59 | 0.65 | 0.70 | 0.74 | **0.75** | 0.58 | 0.64 | 0.69 | 0.73 | **0.74** |
| **ResNet50 / CIFAR-100** | 0.55 | 0.60 | 0.65 | 0.70 | **0.71** | 0.52 | 0.57 | 0.61 | 0.67 | **0.68** | 0.48 | 0.53 | 0.58 | 0.63 | **0.64** | 0.49 | 0.54 | 0.59 | 0.64 | **0.65** |
| **BERT-base / 20News** | 0.70 | 0.75 | 0.79 | 0.82 | **0.83** | 0.70 | 0.75 | 0.79 | 0.83 | **0.84** | 0.69 | 0.74 | 0.78 | 0.82 | **0.83** | 0.70 | 0.75 | 0.79 | 0.83 | **0.84** |

(often $\leq 0.3$ pp) under the same budget. Eight local epochs are *enough*: they capture the client distribution well, while additional epochs yield diminishing returns; fewer epochs underfit and fail to expose sufficient salience structure for clustering to exploit. In contrast, under-training markedly hurts: 2 epochs are about 10–15 pp worse than 8 depending on model/regime, and 4 epochs lag by 6–10 pp. The effect is most pronounced in harder regimes (Dir(0.1), Patho) and for deeper networks (ResNet50), where more local steps are needed to shape client-specific heads and stabilize overlap. On *ResNet50/CIFAR-100* with Dir(0.1), accuracy climbs from 35.10 (2 ep) to 39.90 (4 ep), 43.40 (6 ep), and 48.30 (8 ep), with only a negligible rise to 48.50 at 10 epochs; similarly, for *BERT-base/20News* under Patho(30%), scores go $44.50 \rightarrow 48.90 \rightarrow 52.10 \rightarrow 54.25$ with a tiny lift to 54.34 at 10 epochs, while *CNN/FMNIST* under Patho(20%) jumps from 83.40 (2 ep) to 95.18 (8 ep) and only nudges to 95.30 at 10.

**Neighbor Selection: Retrieval Quality vs. Oracle.** We build the similarity graph with *IOS* at $K^\star$ for each epoch setting and perform $k$-NN ($k=8$) neighbor selection. We then compute Recall@8 against oracle neighbors (defined via Wasserstein distance over true per-client label histograms; used only for evaluation).

Table 11 reports Recall@8 across epochs $2, 4, 6, 8, 10$. Neighbor retrieval improves smoothly with more local training and stabilizes by 8 epochs; moving to 10 brings at most a $+0.01$ gain in Recall@8. Under-training substantially lowers recall: at 2 epochs we see about 0.10–0.15 absolute drops relative to 8, at 4 epochs about 0.06–0.10, and at 6 epochs about 0.01–0.07. The pattern mirrors accuracy: deeper models and stronger heterogeneity require more local updates before the importance head is sharp enough to recover oracle-like neighborhoods. For *ResNet50/CIFAR-100* (Dir(0.1)), Recall@8 increases from 0.52 (2 ep) to 0.57 (4 ep), 0.61 (6 ep), and 0.67 (8 ep), with only a minimal change to 0.68 at 10; on *ResNet18/CIFAR-10* with Patho(20%), the trajectory $0.59 \rightarrow 0.65 \rightarrow 0.70 \rightarrow 0.74 \rightarrow 0.75$ exhibits the same saturation; and *BERT-base/20News* under Dir(0.3) moves from 0.70 (2 ep) to 0.82 (8 ep) with a tiny step to 0.83 at 10.

## F USE OF LARGE LANGUAGE MODELS (LLMs)

**Scope and intent.** LLMs were used *only* to aid and polish writing (grammar, clarity, concision, tone, and LaTeX hygiene). They were *not* used to design experiments, analyze data, generate results, choose hyperparameters, or create figures/tables. All technical contributions, algorithms, proofs, and empirical results originate from the authors.

**Tools.** We used ChatGPT (GPT-5 Thinking) in an editorial capacity. Typical operations included: sentence rephrasing for clarity, reducing redundancy, harmonizing terminology/notation, improving caption phrasing, fixing cross-references, and standardizing style (e.g., capitalization, punctuation, hyphenation). When requested, it proposed concise alternatives that the authors reviewed and edited.

**Content provenance and verification.** No passages were accepted verbatim without author review. The model was *not* allowed to introduce new claims, citations, equations, or numbers. All references and quantitative values in the paper were produced by our code/experiments and cross-checked by the authors.

**Data and privacy.** We did *not* upload raw datasets, private code, or proprietary results. Shared text was limited to draft paragraphs, captions, and LaTeX snippets necessary for stylistic edits. No confidential or personal data were provided to the LLM.

**Bias and accountability.** LLMs may reflect stylistic or cultural biases. Final wording, framing, and interpretations are the authors' responsibility. Any errors remain our own.

**Reproducibility note.** The use of LLMs does not affect the reproducibility of results. All experiments can be reproduced from the released code, configurations, and seeds; LLM involvement was purely editorial.

**Authorship.** All authors reviewed and approved the final text. The LLM is not listed as an author and did not meet authorship criteria.

