# OpenReview forum: "Index-Overlap Similarity: A Value-Free Proxy for Model Relatedness"
_ICLR.cc/2026/Conference — ICLR 2026 Conference Desk Rejected Submission_

### Official Review · Reviewer_okFg · 2025-10-24

**Soundness:** 3
**Presentation:** 3
**Contribution:** 3
**Rating:** 4
**Confidence:** 3

**Summary:**

This paper tackles the problem of measuring model relatedness in federated learning without incurring high communication cost or risking privacy leakage from sharing real valued parameters. It **introduces Index Overlap Similarity (IOS)**, a value-free metric that represents each model by the **indices** of its **Top $K$ most important parameters** and measures similarity through their normalized overlap. The authors provide theoretical guarantees showing that IOS lower bounds cosine similarity (Proposition 1) and that the Top $K$ support remains stable under noise when smoothed with an EMA (Theorem 1). Experiments on FMNIST, CIFAR10, CIFAR100, and 20News show that IOS matches or outperforms cosine and Euclidean metrics across four federated learning tasks: clustered PFL, neighbor selection, donor ranking, and oracle distribution alignment.

**Strengths:**

**1) Strong Empirical Validation:** IOS is tested as a drop-in replacement for cosine similarity across four federated learning tasks: clustered PFL, neighbor selection, donor ranking, and oracle distribution alignment. It shows consistent gains on both vision and text tasks using diverse models demonstrating strong robustness and generality.

**2) Significance of the Value-Free Design:** The value-free, index-only idea behind IOS is simple and important. By using only the indices of the most important parameters instead of their values, IOS offers a practical alternative when sharing real-valued updates is costly or raises privacy concerns.

**3) Methodological Completeness and Practicality:** The paper presents a complete and well designed method. The procedure for selecting $K$ is principled and local, balancing coverage $(\tau) $ and stability $ (\rho_0)$ instead of treating $K$ as fixed (Algorithm 1).

**4) Clear Motivation and Writing:** The paper is clearly written and well-organised. The introduction frames the work around three main challenges of communication cost, privacy risk, and numerical instability, making the motivation and importance easy to understand. That being said, I later raised a question in my review about how fully these motivations are addressed.

**Weaknesses:**

**1) Reproducibility and Missing Code (Major):**
The paper’s contribution relies heavily on experimental validation, comparing the effectiveness of IOS across several federated learning tasks. However, a link to the code is not provided in the paper or the supplementary material, making it difficult for others to reproduce the results or verify the findings. Since the work is largely experiment based, providing access to the code is important for supporting the credibility and reliability of the results.

**2) Missing Evaluation of Privacy and Communication Benefits:**
The paper highlights privacy and communication efficiency as key motivations for IOS (first paragraph of the introdcution), yet the experiments focus only on performance and similarity. There is no quantitative analysis of communication savings or empirical evaluation of privacy protection, leaving these claims unverified. Moreover, although the authors note that they “make no formal privacy claims,” this is a major gap, as sharing the indices of the Top-$K$ most salient parameters could still reveal sensitive information about client data. For example, an attacker might infer aspects of a client’s data distribution if certain classes consistently activate specific parameters.

**Minor Point:** In the appendix, Section E appears to be empty. It seems that the following section (Section F) was intended to be a subsection of it.

**Questions:**

**1.** Theorem 1 assumes independent, mean-zero sub-Gaussian coordinates. How realistic is this assumption for gradient-based importance measures in deep networks? Can these measures not often be heavy-tailed and correlated across layers? More broadly, could the stability theorem be extended to cases with sub-exponential or heavy-tailed noise, which are frequently observed in deep learning settings?

**2.** Could the authors provide at least a preliminary analysis or discussion of potential information leakage risks (e.g., linkage or reconstruction attacks) that might arise from sharing Top-$K$ indices, and how such risks could be mitigated?

**3.** In line 697 of the proof of Proposition 1, could the authors clarify the use of $s_{ij} $? Should this term be replaced by $ |I_i \cap I_j| $ so that dividing by $K$ yields $R_{ij} $, consistent with the final statement of the proposition?

---

> ### Author Response · Authors · 2025-11-20
> **Response to Reviewer okFg**
>
> ### Weaknesses
> * **W1: Reproducibility and missing code.**
> We agree that code availability is important for a largely experimental paper and apologize for the confusion. The ICLR author guidelines describe three options for sharing code; we followed the third option (posting an anonymous repository link in a comment to reviewers and ACs after discussion opens), which is why the initial PDF did not contain a GitHub URL. The full implementation, configurations, and scripts are already available on **[GitHub in an anonymized repositories](https://github.com/Index-Overlap-Similarity)**, and we will make this repo public upon acceptance. We hope this clarifies our choice and alleviates concerns about reproducibility.
> * **W2: Missing evaluation of privacy and communication benefits.**
> We have now added both privacy and communication analyses. Sec. 2.2 clarifies the honest-but-curious threat model and that IOS never exposes raw gradients/activations, so **standard reconstruction attacks do not apply directly**. Sec 6.3 adds a **membership-inference experiment** where the adversary observes either (i) full gradients or (ii) only IOS index sets: across FMNIST, CIFAR-10, CIFAR-100, and 20NewsGroups, attack accuracy drops from $\approx$80--90\% with gradients to $\approx$50--70\% with indices (50\% is random guessing), indicating substantially lower leakage for IOS. We also provide a quantitative communication study in Appx E.1 (Table 7), reporting bits per client and the corresponding uplink time at 500\,Mb/s, and show 8--14$\times$ payload reductions for the index-only channel at our $K/M$ ratios. We present IOS as reducing the attack surface and communication cost relative to value-sharing baselines, rather than as offering a formal privacy guarantee.
> ### Questions
> * **Q1. Sub-Gaussian assumption and heavy-tailed noise.**
> We agree that strict, independent sub-Gaussian coordinates are idealized. In the revised version, we add a remark related to Theorem 1 (Appx C.4), clarifying that the proof only needs coordinate-wise exponential tails and temporal independence; the argument extends to sub-exponential noise via a Bernstein-type bound with a modified variance proxy and an extra logarithmic factor. Truly heavy-tailed and strongly correlated settings would require either stronger margin conditions or robustified estimators, which we explicitly leave as future work.
> * **Q2. Information leakage from Top-$K$ indices.**
> As explained earlier, Sec. 2.2 now discusses the threat model and why IOS does not expose raw gradients/activations, making standard reconstruction attacks less applicable. In Sec. 6.3, we add a preliminary membership-inference study where the adversary observes either gradients or only Top-$K$ indices; the latter yields much lower attack accuracy (close to chance), and we outline that IOS mitigates the reconstruction attack by avoiding raw gradient exchange.
> * **Q3. Clarification in the proof of Proposition 1.**
> Thank you for catching this. We have replaced the ambiguous term at the indicated line with the intersection form $|I_i \cap I_j|$ so that, after dividing by $K$, the expression is consistent with the final statement in terms of $R_{ij} = |I_i \cap I_j|/K$.

---

### Official Review · Reviewer_xNXp · 2025-10-31

**Soundness:** 2
**Presentation:** 1
**Contribution:** 3
**Rating:** 4
**Confidence:** 3

**Summary:**

The paper proposes a novel model similarity metric, Index-Overlap Similarity (IOS), for Federated Learning (FL). It assumes a standard FL setup where clients begin with a global model and train it on their local data. The work is motivated by the communication overhead, privacy risks, and numeric instability associated with traditional "value-based" similarity metrics.

The proposed method requires clients to locally determine the importance of their model's parameters (using a diagonal Fisher proxy) and then transmit only the indices of the Top-K most important parameters. The similarity between any two clients is then calculated as the normalized overlap of these two index sets.

To validate this approach, the paper provides theoretical justifications for the metric's stability and its connection to cosine similarity. It also presents an empirical evaluation across four distinct FL applications (e.g., Clustered FL, Neighbor Selection), comparing IOS against Cosine and Euclidean distance as baselines.

**Strengths:**

+ The paper addresses a well-defined and critical problem in FL. Finding a communication-efficient, robust, and privacy-preserving proxy for client-relatedness is a significant challenge.

+ The core idea of using the support (the set of salient indices) rather than the parameter values is intuitive and an elegant way to abstract model-level relatedness.

+ Taken at face value, the empirical results are strong. The IOS method consistently outperforms the Cosine and Euclidean baselines across all four applications, often by a noticeable margin.

+ The choice of four different experimental applications (CPFL, Neighbor Selection, Donor Ranking, and Distribution Alignment) is commendable and provides a comprehensive view of the method's potential utility.

**Weaknesses:**

- The paper dedicates its introduction to explaining why Cosine and Euclidean distance are flawed, unstable, and "debilitating" metrics, yet these are the only baselines it compares against. A convincing evaluation must include comparisons against modern, robust model similarity metrics. The paper explicitly cites methods such as CKA (Central Kernel Alignment) and FedProto, but makes no effort to compare against them or justify their exclusion. This use of "strawman" baselines makes it impossible to assess the true contribution of IOS.

- A primary motivation for IOS is its "value-free" nature, which the paper claims "advancing scalable personalization in decentralized learning" with a "strong privacy posture". However, this claim is not supported by any formal privacy proof (e.g., differential privacy) or empirical leakage analysis. It is not self-evident that leaking the Top-K indices of the most important parameters is inherently more private than, for example, sharing the penultimate layer activations for a CKA calculation. The privacy benefits are asserted, not proven.

- The paper is also motivated by efficiency and low overhead. However, it fails to provide a proper analysis of the new costs it introduces. The method requires new, non-trivial client-side computation: estimating the diagonal Fisher proxy and performing a global Top-K selection over all model parameters. The paper provides no analysis (neither theoretical nor empirical) of this computational trade-off, making its efficiency claims one-sided.

- The efficiency and communication arguments are motivated by the challenge of "multi-billion-parameter checkpoints". However, the experiments are conducted on much smaller models, with ResNet-50 and BERT-base being the largest. The feasibility of performing a global Top-K selection on a vector with billions of entries on a client device is not discussed, creating a significant gap between the paper's motivation and its evidence.

- The paper feels very unpolished and is riddled with minor errors. For example, in Table 4, IOS is italicized for FMNIST, but bold-faced for other datasets. There are multiple cases in Table 3 where the results for the proposed methodology are in bold font instead of better results attained by the baselines. Table formatting is inconsistent.

**Questions:**

Why did the evaluation exclude comparisons against contemporary and potentially more robust model-relatedness metrics, such as CKA or FedProto, both of which the paper cites?

Given the assertion of a strong privacy posture, could the paper provide an empirical analysis quantifying the information leakage (e.g., via model inversion or membership inference attacks) from the shared Top-K index sets compared to the leakage from the baseline real-valued vectors?

Please clarify the explicit criterion for determining the best result (i.e., the bolded number) across all tables, as inconsistencies exist (e.g., in Table 3). Furthermore, were the observed empirical gains over the baselines subjected to tests for statistical significance? If so, please provide the relevant statistical evidence (e.g., confidence intervals or $p$-values) to confirm.

---

> ### Author Response · Authors · 2025-11-20
> **Response to Reviewer xNXp**
>
> ### Weaknesses
> * **W1: Stronger baselines: CKA, FedProto, and FedBary.**
> We agree that CKA and FedProto are important references, but they are not directly comparable to IOS in our setting: both operate on **value-bearing representations** (activations/prototypes), require a shared probe dataset or shared prototypes between clients and server that is not present in most FL settings, and ultimately rely on cosine/Euclidean geometry over real-valued vectors. Our threat model assumes no shared data and a value-free interface, so these methods fall outside the scope of what IOS is designed to replace; we now state this explicitly in Sec 1. To address the spirit of the comment, Sec. 6.2 adds a comparison with a **Wasserstein-barycenter metric (FedBary)**. Also, it should be noted that our aim is not to prove IOS strictly dominates cosine/Euclidean, but to show it is *comparable* to these widely used baselines while avoiding real-valued channels, reducing communication, and enabling personalization even under HE/SMPC/DP protections.
> * **W2: Privacy claims and "value-free" benefit.**
> First, note that standard reconstruction / gradient-inversion attacks in FL critically rely on access to raw gradients or activations; IOS never transmits these, only index sets, so such attacks are not applicable in our setting. That said, we agree that "value-free" does not automatically imply "privacy-free", and we thank the reviewer for pushing us to quantify this aspect. To provide evidence beyond intuition, Sec 6.3 adds a concrete **membership-inference experiment** in which the adversary observes either (i) full gradients or (ii) only IOS index sets. Across our models, attack accuracy drops from high values (e.g., $\approx$80--90\%) under gradients to $\approx$50--70\% under indices (with $50\%$ being random guessing), indicating substantially lower leakage from IOS than from gradient sharing.
> * **W3: Efficiency and client-side overhead.**
> We agree that our original submission did not sufficiently quantify the new costs introduced by IOS. Sec. 4.4 and Appx. D.4 now provides a simple **theoretical cost model** and analyzes the ratio $C_{IOS}/C_{train}$, showing that both the EMA update and the Top-$K$ selection are linear-time vector operations whose cost is dominated by training. To complement this, Appx. E.1 (Table 6) reports **wall-clock measurements** on all models, with an ablation that times (i) Fisher/EMA alone, (ii) Top-$K$ selection alone, and (iii) their combination. Across all settings, IOS instrumentation adds only $\approx$2.5--4\% to the total local training time. For communication, Appx. E.1 (Table 7) reports the number of bits sent per client and payloads for the index-only channel at the $K/M$ ratios we use.
> * **W4: Multi-billion-parameter motivation vs. current models.**
> We agree with your comment. However, our goal there was to motivate the regime where value-based similarity is especially costly, not to claim that our experiments have already run at that scale; the model sizes we use are in line with most empirical FL work. Algorithmically, IOS remains $O(M)$ in model size: the diagonal proxy and global Top-$K$ are single passes, using the same primitives as standard methods in large-scale FL. Sec. 4.4 Discuss a large-$M$ scenario together with MinHash--LSH design, and we explicitly note that a full billion-parameter LLM evaluation is an important direction for future work rather than part of the current experimental scope.
> * **W5: Presentation issues and table formatting.**
> We thank the reviewer for carefully catching these issues. The inconsistent use of formating in Tables 3 and 4 and the table formatting problems have now been corrected.
> ### Questions
> * **Q1. Why no CKA/FedProto baselines.**
> As noted above, CKA and FedProto operate on value-bearing representations, require a shared probe dataset or shared prototypes, so they are outside the class of methods IOS is meant to replace. We now state this explicitly in Sec. 1 and, to address the spirit of the comment, add a Wasserstein-barycenter CPFL (FedBary) baseline in Sec. 6.2.
> * **Q2. Empirical leakage from Top-$K$ indices.**
> Yes. Sec. 6.3 now reports a membership-inference experiment where the adversary observes either (i) full gradients or (ii) only IOS index sets. Across all models, accuracy drops from $\approx$80--90\% with gradients to $\approx$50--70\% with indices (50\% random guessing), showing substantially lower leakage for IOS.
> * **Q3. Boldface criterion and statistical significance.**
> We now use a single rule: boldface always denotes the best mean performance in each block, and the formatting issues in Tables 3/4 have been fixed. We did not run formal statistical tests; all means are averaged over multiple seeds. Our aim is not to prove IOS strictly dominates cosine/Euclidean, but to show it is *comparable* to these baselines while avoiding valued transmission, reducing communication, and personalization even under HE/SMPC/DP protections.

---

### Official Review · Reviewer_gaMV · 2025-10-31

**Soundness:** 2
**Presentation:** 2
**Contribution:** 2
**Rating:** 4
**Confidence:** 4

**Summary:**

The paper proposes a novel, value-free metric called Index-Overlap Similarity (IOS) to quantify the similarity between client models in a Federated Learning environment. The core idea is to move away from transmitting high-dimensional, real-valued weights or gradients. Instead, each client's model is represented by an index set (I_i) (or "support set") of its Top-K most significant parameters. The similarity is then computed solely based on the normalized overlap of these index sets. This approach claims to significantly reduce communication bandwidth and narrow the privacy attack surface created by the leakage of real-valued parameters.

**Strengths:**

1. Experiments are thorough, covering both vision and text domains (FMNIST, CIFAR-10/100, 20News) and different application scenarios.
2. The work is not purely empirical, offering critical theoretical insights, particularly Proposition 1 (Lower Bound for Cosine Similarity) and Theorem 1 (Stability of Top-K Selection). This greatly enhances the credibility and explainability of the method.
3. The paper addresses the similarity calculation overhead for a large number of clients by mentioning the use of MinHash-LSH, demonstrating attention to system-level scalability.

**Weaknesses:**

1. Privacy is a core motiviation, but the Top-K index set I_i itself encodes sensitive information about data distribution and model structure. The paper lacks in-depth analysis and quantification of the risk associated with index leakage, which is a major weakness.
2. Certain parameters need clearer definitions, for example, M (total number of model parameters) and the meaning of C(K) (proportion of total importance represented by the Top-K parameters).
3. IOS performance is highly dependent on the optimal support set size K^* chosen by Algorithm 1. Even small increases in K (e.g., 15% or 25%) lead to a significant drop in downstream task performance. This raises concerns about the robustness of the automatic K selection mechanism
4. The paper fails to clearly explain the method for model parameter upload and aggregation in the CPFL context, only providing a client similarity metric.

**Questions:**

1. Given that privacy is a core motivation, could you more deeply discuss the specific attack vectors that could arise from only leaking the index set I_i? For instance, what inferences could an adversary make about the model structure or the underlying data distribution based on the indices of the most significant parameters? A quantified analysis of this index-leakage risk is requested.
2. The main benefits of IOS are on the server side and in communication. Please quantify the absolute time overhead introduced by IOS on the client side. Specifically, compared to the total time for one local SGD and backpropagation, what is the proportional increase in wall-clock time spent on computing the Fisher diagonal proxy and Top-K selection? Preferably with specific ablation experiments.
3. The paper mentions using a mechanism similar to IFCA (or similar approaches) to train the clustered models. Could the authors explicitly explain how the parameter fusion/aggregation is actually performed? Does the system still involve directly sharing the numerical values of the model parameters for the aggregation step, or is the information transfer still restricted to the "value-free" indices even during the final model update?

---

> ### Author Response · Authors · 2025-11-20
> **Response to Reviewer gaMV**
>
> ### Weaknesses
>
> * **W1: Privacy and index leakage.**
> We agree that value-free does not mean privacy-free, and we thank the reviewer for pushing us to quantify this aspect. In Sec. 2.2 ("Threat model and scope'') we now explicitly state an honest-but-curious coordinator and clarify that in applications which already share model parameters (e.g. CPFL), these are protected by HE/partial-HE that enable aggregation on encrypted models or use DP to add noise to the gradient. given that calculating similarity on encrypted data is not practical and similarity is unstable on DP, IOS only adds index sets on top. In section 6.3, we add a **concrete membership-inference experiment** where the adversary observes either full gradients or only IOS index sets. Across FMNIST, CIFAR-10, CIFAR-100, and 20NewsGroups, attack accuracy drops from high values (e.g., $\approx$80--90\%) under gradients to $\approx$50--70\% under indices (with $50\%$ being random guessing), indicating substantially lower leakage from IOS than from gradient sharing.
>
> * **W2: Definition of $M$ and $C(K)$.**
> Thank you for catching this. In Sec. 3.1, We now define $M$ at first use as the number of scalar parameters in the model. In Sec. 3.2 and Appx D.2 we clarify that $C_i(K)$ is the fraction of total importance mass captured by the Top-$K$ coordinates of client $i$ ("cumulative importance coverage''). We use this interpretation consistently in the $K^\star$ selection rule.
>
> * **W3: Sensitivity to $K^\star$**
> We agree that $K$ is the most important hyperparameter in IOS and that robustness to its choice is crucial. In fact, a key part of IOS's novelty is precisely that **we do not treat $K$ as a predefined or end-user–chosen global knob**: Algorithm 1 performs a local binary search over $K$ on each client to balance cumulative importance coverage $C_i(K)$ and temporal stability, thereby **selecting a data-driven $K^\star$ rather than relying on ad hoc heuristics**.
>
> * **W4: CPFL upload and aggregation pipeline.**
> We acknowledge the lack of clarity in the previous version and thank the reviewer for pointing this out. The new Sec. 3.3 ("Applying IOS in FL Applications'') now explicitly describes the full pipeline. We first distinguish between (i) similarity-only applications of IOS (e.g., neighbor selection, donor ranking, oracle alignment), where only index sets are exchanged, and (ii) applications that still require parameter aggregation, such as CPFL. For the latter class, our threat model assumes that value-bearing updates are already protected by HE/partial-HE, SMPC, or DP. However, these mechanisms make it impractical or inaccurate to compute similarities directly on encrypted or DP-noisy vectors. IOS removes this barrier by requiring only the additional upload of local Top-$K$ index sets $I_i$ alongside the protected gradients: personalization decisions (e.g., cluster assignment) are driven by index-overlap similarity, while aggregation itself continues to operate on shielded numerical updates.
>
> ### Questions
>
> * **Q1. Attack vectors and quantified index-leakage risk.**
> We now discuss possible inferences and provide a quantitative experiment. Sec. 6.3 elaborates that an adversary observing Top-$K$ indices could, in principle, infer coarse information such as which layers or filters are most salient or whether certain classes consistently activate specific regions of the network. Sec. 6.3 then instantiates a membership-inference attacker that receives (a) full gradients vs. (b) only IOS index sets.
>
> * **Q2. Client-side overhead of Fisher and Top-$K$.**
> We added both analytic and empirical overhead analyses. Sec. 4.4 and Appx. D.4 derives the cost of IOS instrumentation, relative to the training cost. Appx E.1 (Table 6) reports **wall-clock measurements** on all models. Across all settings, Fisher+ Top-$K$ adds only $\approx$2.5--4\% to the total local training time, with both the EMA update and the Top-$K$ selection implemented as simple linear-time vector operations. We also break this down into the importance-estimation and Top-$K$ phases, and show absolute times, confirming that client-side overhead is modest. A similar analysis for communication time is provided in Table 7.
>
> * **Q3. IFCA-style clustering and aggregation: values vs. indices.**
> Sec. 3.3 now makes this explicit. Our CPFL implementation follows an IFCA-style protocol in which IOS only replaces cosine in the clustering step. Clients still upload shielded numerical updates (e.g., gradients or parameter deltas), which are averaged within each cluster, while index sets $I_i$ are used solely for cluster assignment via IOS. Thus, IOS provides a value-free similarity signal but does not remove the need for value-based aggregation; in privacy-preserving deployments, these values are protected by HE/partial-HE or DP as described in Sec. 3.3.

---

### Author Response · Authors · 2025-11-20
**Revision and generic comment**

We thank all reviewers for their careful reading and constructive feedback. We are encouraged that all three reviewers appreciated the core idea of IOS, the theoretical development, the novelty of the framework, and the breadth of applications, and that the main concerns focus on missing analyses and clarifications rather than flaws in the approach itself.

Across the reviews, the key issues cluster into three aspects:
(1) privacy, Robustness against attacks, and potential leakage from Top-$K$ indices;
(2) computational and communication overhead, and the ablation study on the overhead of different modules of IOS;
(3) positioning and completeness of the evaluation, including stronger baselines, CPFL aggregation details, and reproducibility.

In response, we have:

  i) Privacy and threat model: clarified the honest-but-curious threat model(Sec. 2.2), and added an **empirical membership-inference study** comparing gradient leakage to IOS index leakage while describing why a **reconstruction attack can not apply to IOS** (Sec. 6.3).

  ii) Overhead analysis: provided analytic **cost expressions in terms of communication and computation time** (Sec. 4.4 and Appx. D.4), and also new wall-clock measurements of client-side training, important calculation, EMA update, and Top-$K$ overhead, quantified communication savings (Appx. E.1)

iii) Positioning, baselines, and clarity: clarified how IOS differs from representation-level metrics (CKA, FedProto, etc.), added a stronger similarity baseline where applicable, detailed the CPFL aggregation pipeline (Sec. 3.3), adding theoretical remarks for going beyond sub-Gaussian noise (Appx. C.4), and supplied an anonymized code link for reproducibility.

We have not changed the core technical content or main claims of the original submission, except for minor edits explicitly requested by the reviewers; instead, we have added new analyses and clarifications, which are highlighted in red in both the main text and appendix. Since IOS provides a general framework for model similarity that can be applied across many machine-learning settings, we kindly ask you to consider these additions when reassessing the paper, and we would be grateful for any further changes you deem necessary. In addition, we provide separate comments for each reviewer, addressing their stated weaknesses and questions individually.

---

### Note · Program_Chairs · 2026-01-17
**Submission Desk Rejected by Program Chairs**

The following references in this submission do not refer to real documents and/or have major errors in bibliographic information:

 X Lin et al. Measuring and learning data quality for fl via shapley values. In AAAI, 2022.
J Wang et al. Measuring the influence of clients in federated learning. In NeurIPS Workshop on FL, 2020.